



# Surface-atmosphere exchange of ammonia over peatland using QCL-based eddy covariance measurements and inferential modeling

Undine Richter[1], Christian Brümmer[1], Frederik Schrader[1], Christof Ammann[2], Andreas Ibrom[3], Christophe R. Flechard[4], David D. Nelson[5], Mark Zahniser[5], Werner L. Kutsch[6]

[1]Thünen Institute of Climate-Smart Agriculture, 38116 Braunschweig, Germany
[2]Swiss Federal Research Station Agroscope ART, CH-8046, Zürich, Switzerland
[3]Technical University of Denmark, Department of Environmental Engineering, Bygningstorvet, 2800 Kgs. Lyngby, Denmark
[4]INRA, Agrocampus Ouest, UMR 1069 SAS, 35042 Rennes, France
[5]Aerodyne Research, Inc., Billerica, Massachusetts, USA
[6]Integrated Carbon Observation System (ICOS), Head Office, University of Helsinki, Finland

*Correspondence to*: Undine Richter (undine.richter@thuenen.de)

**Abstract.** Recent advances in laser spectrometry offer new opportunities to investigate ecosystem-atmosphere exchange of environmentally relevant trace gases. In this study, we demonstrate the applicability of a quantum cascade laser (QCL) absorption spectrometer to continuously measure ammonia concentrations at a high time resolution and thus to quantify the net exchange between a semi-natural peatland ecosystem and the atmosphere based on the eddy-covariance approach. Changing diurnal patterns of both ammonia concentration and fluxes were found during different periods of the campaign. We observed a clear tipping point in early spring with decreasing deposition velocities and increasingly bi-directional fluxes that occurred after the switch from dormant vegetation to $CO_2$ uptake, but was triggered by a significant weather change. While several biophysical parameters such as temperature, radiation, and surface wetness were identified to partially regulate ammonia exchange at the site, the seasonal concentration pattern was clearly dominated by agricultural practices in the surrounding area. Comparing the results of a compensation point model with our measurement-based flux estimates showed considerable differences in some periods of the campaign due to overestimation of non-stomatal resistances caused by low acid ratios. The total cumulative campaign exchange of ammonia after nine weeks, however, differed only in a 6 % deviation with 911 and 857 g NH3-N ha$^{-1}$ deposition being found by measurements and modeling, respectively. Extrapolating our findings to an entire year, ammonia deposition was lower than reported by Hurkuck et al. (2014) for the same site in previous years using denuder systems. This was likely due to a better representation of the emission component in the net signal of eddy-covariance fluxes as well as better adapted site-specific parameters in the model. Our study not only stresses the importance of high-quality measurements for studying and assessing land surface-atmosphere interactions, but also demonstrates the potential of QCL spectrometers for continuous observation of reactive nitrogen species as important additional instruments within long-term monitoring research infrastructures such as ICOS or NEON.



## 1 Introduction

Increased agricultural production and energy consumption over the last century led to a dramatic increase in anthropogenic reactive nitrogen (N) production (Erisman et al., 2008). Atmospheric N deposition can be a major driver of change in most natural and semi-natural ecosystems and may considerably alter species composition, biodiversity, and ecosystem functioning with regard to causing nutrient imbalances. As ammonia ($NH_3$) mainly originates from agricultural activities, it has received more and more attention in the past 20 years (e.g., Sutton et al., 2011). It is estimated that in 2008 65 Mt $NH_3$-N were emitted globally to the atmosphere (Sutton et al., 2013), a large proportion (60 %) of it from anthropogenic sources. Around 1910 the creation and therefore release of $NH_3$ experienced a steep rise due to the Haber-Bosch-process (Galloway et al., 2003). While ammonia emissions doubled since 1950 (Asman et al., 1998), Sutton et al. (2013) predict the emission of 132 Mt $NH_3$-N $yr^{-1}$ by 2100. Ammonia is an essential part of the nitrogen cascade with reactive nitrogen ($N_r$) tending to accumulate in both, ecosystems and atmosphere, mainly because denitrification in soils cannot balance out industrial N creation (Galloway et al., 2003) causing a severe environmental problem for future generations. Additionally, reactive N is widely spread through hydrologic and atmospheric transport processes (Galloway et al., 2003). Ammonia can cause foliar injury, increase sensitivity to drought, reduce frost hardiness, lead to structure change in plant communities and loss in biodiversity, especially regarding sensitive nutrient-poor ecosystems like peatbogs (Krupa, 2003). Furthermore, deposited ammonia causes ecosystem acidification, fertilization, and eutrophication (Galloway et al., 2003), while air and water quality deterioration also impact human health (Erisman et al., 2013).

Until now only little is known about the temporal and spatial variability of $NH_3$ concentrations and exchange fluxes between different ecosystems and the atmosphere. Estimates to what extent $NH_3$ is being emitted from plant canopies under common environmental conditions remain highly uncertain (e.g., Flechard et al., 2013). This is due to the fact that continuous half-hourly, micrometeorological measurements of $NH_3$ exchange, e.g. based on the aerodynamic gradient or eddy-covariance (EC) technique, have largely remained experimental and were limited to selected research sites and to measurement campaigns of typically a few weeks to a few months due to technical complexity and to the large equipment and operational costs involved (e.g., Sutton et al., 2007; Flechard et al., 2011; Marx et al., 2012; Brümmer et al., 2013). Other well-established techniques like denuder, impinger or filter sampling in combination with ion chromatography analysis usually provide concentration values and flux rates at poor time resolution and require labor- and cost-intensive chemical analyses (e.g., Dämmgen and Zimmerling, 2002; Tang et al., 2009; Hurkuck et al., 2014). Over the last few years, substantial progress has been made in the use of tunable diode laser absorption spectrometers (TDLAS) and quantum cascade lasers (QCL). The precision and fast response of these approaches have allowed first EC measurements of field scale $N_2O$ and $CH_4$ fluxes (Denmead et al., 2010; Kroon et al., 2010; Neftel et al., 2010; Merbold et al., 2014) and are expected to become a standard method within integrated observation networks such as ICOS in Europe or NEON in North America. Eddy-covariance measurements of $NH_3$ fluxes, however, have been extremely limited and are still subject to considerable uncertainty (Famulari et al., 2004; Ellis et al., 2010; Sintermann et al., 2011; Ferrara et al., 2012) mainly due to issues regarding the tube





inlet design, sampling high frequency flux losses and tube wall and air chemistry (e.g., Marx et al., 2012 and references therein).

In this study, we demonstrate the applicability of a quantum cascade laser (QCL) absorption spectrometer to continuously and fast measure turbulent fluctuations at background-level $NH_3$ concentrations and thus the net exchange between a semi-natural peatland ecosystem and the atmosphere, with the eddy-covariance technique. Specifically, we (1) test the QCL performance to measure $NH_3$ concentration fluctuations and calculate $NH_3$ fluxes and net deposition during the observation period, (2) investigate the biophysical controls on $NH_3$ concentrations and fluxes, and (3) compare the measured fluxes with results from a local application of a two-layer $NH_3$ canopy compensation point model using the parameterization after Massad et al. (2010). The general objective is to better understand the mechanisms of peatland-atmosphere $NH_3$ exchange under the influence of highly intensive agricultural land management in the surrounding area.

## 2 Materials and methods

### 2.1 Site description and local climate

Tower-based fast response measurements of ammonia concentrations were conducted from 18 February to 8 May 2014 at an ombrotrophic, moderately drained peatland located in Northwest Germany near the city of Meppen (52°39'19.98"N, 7°10'56.65"E, 14 m.a.s.l.). The site is surrounded by intensive agricultural land and livestock holdings. Local vegetation is dominated by bog heather (Erica tetralix), purple moor-grass (M. caerulea), cotton grass (Eriophorum vaginatum, Eriophorum angustifolium) and is further characterized by some smaller, mostly solitary trees like birches (B. pubescens), and Scots pines (Pinus sylvestris). The peat layer depth is approx. 4 m. The area around the tower has a fetch of 650 m including some small paths, tree lines and hedgerows. Facing towards the main wind direction (SSW) 230 m can be regarded as totally homogenous. Details on fetch and footprint analysis are given in Hurkuck et al. (2014).

From 1981 to 2010 the annual average air temperature ($T_a$) and the mean annual precipitation in the area were 10.0 °C and 800 mm, respectively (German Weather Service, station Lingen, 2015). For the respective months of observation (February to May), the long-term average air temperatures were 2.7 °C, 5.7 °C, 9.3 °C, and 13.6 °C, respectively. Except for May, monthly mean air temperatures in 2014 were higher (4.1 °C, 6.3 °C, 10.9 °C, and 12.3 °C) than the 30-yr averages. Monthly precipitation (36 mm, 31 mm, 39 mm, and 124 mm) was lower in February, March and April than the long-term observations (53 mm, 67 mm, 45 mm and 58 mm). As most of the precipitation was recorded after the campaign on 8 May, the measurement period was considerably drier than the long-term mean.

### 2.2 Measurements of ammonia and microclimate

Ammonia concentrations were measured at high temporal resolution with a QCL absorption spectrometer (model mini QC-TILDAS-76) from Aerodyne Research, Inc. (ARI, Billerica, MA, USA). Laser detectors were thermoelectrically cooled at 25 °C. A 76-m path length and 0.5 L multiple pass absorption cell for sampling at an operation pressure of approx. 40 Torr



was used. The QCL is able to offer up to 10 Hz resolution with the detection limit in the sub-ppb range and a precision of 0.04 ppb when averaged over 1 s (McManus et al., 2008). Along with its compact design, the QCL forms an appropriate basis for eddy-covariance measurements in the field (Ellis et al., 2010, Ferrara et al., 2012). A dry vacuum scroll pump (TriScroll 600, Agilent Technologies, Santa Clara, United States) was used to transport the sample air with a flow rate of approx. 17 l min$^{-1}$ through 3-m long tubing (1 cm inner diameter) to the laser cell and back outside through an exhaust device. For further conceptual details see McManus et al. (2008) and Zahniser et al. (2005) as well as Ellis et al. (2010) for a performance test.

To prevent damage to the laser cell from particles, it is necessary to filter the ambient air. Because of the stickiness of $NH_3$ and its high reactivity, it is not possible to use any conventional membrane filter. Aerodyne Research, Inc. developed a specific inertial inlet (Fig. 1) that removes particles with an aerodynamic diameter larger than 300 nm and that reduces pressure inside tubing and laser cell. After a short PTFE inlet, a critical orifice made of glass ensures that the pressure in the system is decreased to a range of approx. 5.3 to 8 kPa, which significantly reduces wall adsorption effects (Warland et al., 2001) and is required for the operation of the laser cell. After passing the critical orifice, the sample air is forced to make a sharp turn, thereby losing 10 % of its volume, going straight to the pump, and approx. 50 % of the particles (> 300 nm) (Ellis et al., 2010; Ferrara et al., 2012; von Bobrutzki et al., 2010). Heating the inertial inlet box (at approx. 40 °C) and the anti-adhesive PFA tubing that leads the sample air to the QCL's analyzer cell minimizes water condensation or absorption (Massman and Ibrom, 2008, Ibrom et al. 2007), thus avoiding interactions of ammonia with surfaces (Walker et al., 2006; Norman et al., 2009; Ellis et al., 2010). Ellis et al. (2010) compared different tubing variants and reported significant improvement when using heated tubing at 40 °C, which reduced unintended interactions to around 10 % at 30 ppb. A side effect is, that at low atmospheric pressures $NH_4NO_3$ aerosols can be volatilized, if not removed by the inertial inlet, and thereby produce $NH_3$. DELTA denuder measurements (details below) showed a $NH_4^+$ to $NH_3$ ratio of 0.17 up to 0.24 during the campaign. Assuming that 100 % of the $NH_4^+$ aerosols passed the inlet and were volatilized, there was on average an overestimation of 21% of $NH_3$ concentration, which is the same rate Walker et al. (2006) found in their investigation and is an acceptable compromise between $NH_3$ loss due to wall sorption effects and ammonia alterations through $NH_4NO_3$ volatilization. This affects not only the concentration but also the flux, because $NH_4NO_3$ particles are also deposited. In this study, we performed calibration with zero air every 8 hours, i.e. at 00:00, 08:00 and 16:00 local time for 60 s per interval (data not shown) as well as internal system calibration through the laser itself by aligning the $NH_3$ absorption peak of the sampled air to the standard of the HITRAN database (Rothman et al., 2009).

An ultrasonic anemometer (model R3, Gill Instruments, Lymington, UK) was installed at 2.5 m above ground. The inertial inlet box was mounted next to it, placing the sample air inlet 40 cm below the center of the sonic anemometer array. To protect the inlet from rain, a tee-inlet was attached, which allowed to feed in calibration gas (see above).

Additional measurements of $NH_3$ concentrations with monthly time resolution were conducted by means of passive samplers (e.g., Dämmgen et al., 2010) and DELTA denuder (**DE**nuder for **L**ong-**T**erm **A**tmospheric sampling, e.g., Tang et al., 2009). A detailed description of the measurement setup of meteorological parameters such as air and soil temperature, radiation





components, precipitation, water table depth as well as the operation of carbon dioxide and water vapor eddy-covariance measurements is presented in Hurkuck et al. (2014, 2016).

### 2.3 Data acquisition, analysis and flux calculation

Sonic anemometer data were recorded by the EddyMeas software, which is an embedded application of the software package
EddySoft (Kolle and Rebmann, 2009). The QCL was controlled by Aerodyne's TDL Wintel software with ammonia concentration data being recorded at a frequency of 10 Hz on the QCL computer. Anemometer and concentration time series needed alignment to a reference timestamp, before the software EddyPro (LI-COR Inc.) could be used to compute half-hourly exchange fluxes. Block averaging and 2-D coordinate rotation were applied.

The time lag, i.e. the residence time of the air sample in the tubing before it reaches the analyzer cell, was determined via
comparing the maximum covariance of vertical wind speed ($w$) and temperature ($T$) with that of $w$ and $NH_3$ concentration. A strong and irregular drift was observed between the system clock of the sonic anemometer's data acquisition computer and the QCL clock. Thus, time series were shifted against each other on a half hourly basis to force a commonly expected relatively stable lag time using a specifically designed R-script (R Core Team, 2012). After a further covariance maximization procedure, a smaller less varying time lag remained. In EddyPro a range of -2 to 4 s was chosen.

The ogive method from Ammann et al. (2006) was applied to empirically investigate the high-frequency damping of $NH_3$ concentration measurements. Cumulative cospectra of $\overline{w'NH_3'}$ were scaled to the corresponding $\overline{w'T'}$ cospectra in the medium-low frequency range (see example ogives in Fig. 2) for quality filtered cases. The resulting relative deviations at the high-frequency end are quantitative measures of the flux damping factor. A median damping factor of 0.67 was found. As no clear dependency of the damping factor on horizontal wind speed nor atmospheric stability could be observed, a constant
correction factor of 1/0.67 was applied to all $NH_3$ flux values as a simplified approach. The random uncertainty of the correction factor was estimated to 15%, but a potential systematic deviation (part of the damping not appropriately detected by the ogive method) cannot be fully excluded.

To guarantee a high level of data quality, fluxes were flagged according to criteria presented in Mauder and Foken (2006). Data from April 11 to 28 were excluded from analysis due to several technical difficulties such as power outages, pump
failure or insufficient temperature control inside the analyzer housing. 33.9 %, 44.3 %, and 10.9 % of data were flagged with grade 0, 1, and 2, respectively. Data with quality flags 0 and 1 were used for further analysis, while flag 2 data were discarded. Because the stationarity tests included in the flagging protocol by Mauder and Foken (2006) might not be applicable for $NH_3$ due to possibly higher variability of the flux over short time scales, flag 1 was generally not discarded. The remaining 10.9 % of data were not used due to insufficient turbulence ($u_* < 0.1$ m s$^{-1}$). The data gaps (except for the lost
data in April) were filled with the mean diurnal variation (MDV, window= +/- 5 days) method (Falge et al., 2001; Moffat et al., 2007). Campaign data were grouped into four periods (Tab. 1) of 9 to 25 days depending on different concentration patterns and meteorology.



## 2.4 Modeling ammonia exchange

A state-of-the-art dry deposition inferential model driven by measured $NH_3$ concentrations and local micrometeorological conditions was applied to assess plausibility of the flux measurements. We used the parameterization of a two-layer canopy compensation point model from Massad et al. (2010), which simplifies to a one-layer big leaf model for unmanaged ecosystems (i.e., no soil layer is considered explicitly below the canopy). Aerodynamic-, quasi-laminar-, and cuticular resistance ($R_a$, $R_b$, and $R_w$, respectively) were parameterized as described in Massad et al. (2010) for semi-natural/moorland vegetation. Stomatal resistance ($R_s$) was modeled using the simple global radiation and temperature dependent formulation of Wesely (1989), since detailed measurements of vegetation characteristics – which are necessary for more complex approaches (e.g., Emberson et al., 2000) – were not available for the study period. $R_s$ and $R_w$ were calculated with temperature and relative humidity at the mean notional height of trace gas exchange, estimated from measurements at the reference height and measured turbulent sensible ($H$) and latent ($LE$) heat fluxes as described in Nemitz et al. (2009).

## 3 Results and discussion

### 3.1 Diurnal and seasonal pattern of ammonia concentrations

On a half-hourly basis, ammonia concentrations ranged from 2 to 85 ppb with short term (10 Hz) maxima reaching up to 110 ppb. The highest values were found in the beginning of March (Period II) and in the beginning of April (end of Period III), which coincided well with the peak of fertilization activities on nearby agricultural fields including the spreading of organic manures from livestock farming (Fig. 3; for details on farming practices see Hurkuck et al., 2014). The base concentration level outside the fertilization periods ranged mostly between 7 and 15 ppb and is well represented by mean values of Periods I and IV.

The highest mean diurnal variability of ammonia concentrations was found in Period II with peak concentrations being observed in the late afternoon (>30 ppb from 5 to 15 March; Fig. 4). Concentrations were lowest at night during the whole observation period. While in Period III the average mean diurnal course exhibited less variability than in Period II, almost stable concentrations on a low level (7 to 12 ppb) were found in Periods I and IV.

The frequency distributions of wind directions and ammonia concentrations for the whole observation period are shown in Fig. 5. The typical main wind direction from the 200 to 280° sector is clearly visible. However, only the lowest concentrations (<15 ppb, average for 10° wind sectors) were observed under southwesterly winds, whereas peak concentrations were found under winds from the east and northeast. This finding is consistent with observations from previous years using denuder systems (*cf*. Hurkuck et al., 2014). The reason for the concentration dependency on wind direction is the land use in close vicinity to the measurement site. While southwest of the tower and outside the protected zone, the area is characterized by active peat cutting, thus no elevated ammonia concentrations can be expected, east and northeast of the tower a number of farm houses with livestock buildings, manure storage areas and adjacent fertilized land



are located in a distance of approx. 2 km. The pattern of Fig. 5 is also revealed in the progression of wind direction and concentrations over time (data not shown). At the beginning of the campaign, i.e. Period I and early Period II, the predominant wind direction was south and southwest while concentrations were relatively low (Fig. 3). Later on, i.e. from late Period II onwards, wind direction was much more variable with sporadic episodes from the northeast sector, when peak

concentrations between 60 and 110 ppb occurred. Despite the fact that frequent winds from the northeast were recorded in Period IV, it was also the time of the lowest concentration levels ($\bar{c}_{NH_3}$ = 8.0 ppb). This is an indication that the main fertilization activities on the adjacent fields had been terminated during that time.

Similarly diurnal variability of ammonia concentrations with peak values during afternoon over a variety of ecosystems has been observed by other authors (e.g., Sutton et al., 2000; Wolff et al., 2010). There are most likely several reasons for the

observed pattern in this study. First and foremost, concentration levels are highly influenced by agricultural activities in the surrounding area. Farmers usually fertilize their land during the day, thereby causing ammonia volatilization, which is then transported and detected at the study site. With a stable nighttime planetary boundary layer, ammonia in the lower atmosphere is likely being deposited causing decreasing concentrations. With no further penetration from higher layers containing higher loads, concentrations remain low from midnight to sunrise (Fig. 4). When temperatures rise, turbulent

mixing of the planetary boundary layer starts and horizontal exchange with higher layers increases, which, consequently, leads to rising ammonia concentrations over the day. On the other hand, continuous sources like mechanically ventilated stables could cause an opposite pattern with the planetary boundary layer acting as a lid and leading to a concentration build up at night. As no information was available of ventilation types of stables, we assume that land applied manure during the day dominates the concentration signal in Fig. 4.

Beside the strong influence of agricultural management on seasonal concentration variability, temperature is usually a substantial driver. Higher temperature indirectly leads to higher $NH_3$ concentrations (Fig. 7), because it is often related to low relative humidity and thus favors ammonia release from the condensed phase towards the gas phase. But increasing air temperature can also be linked to lower $NH_3$ concentrations, especially observed in this study in late April and early May (Period IV). With higher temperature a larger amount of acidic gas or particle species are present in the atmosphere and

usually leads to reactions of ammonia to ammonium salts such as ammonium nitrate, part of which might even be recorded because of re-volatilization inside the higher temperature and lower pressure in the sampling line (Kim et al., 2011; Norman et al., 2009). We observed higher $NH_3$ concentrations when it was dry and/or cold, whereas rainy conditions led to lower ammonia concentration levels, which confirms findings reported by Mosquera et al. (2001). In our study, high concentrations were at the same time triggered by local sources northeast of the tower as described above. This phenomenon was

particularly observed in the second half of Period III under decreasing air temperatures.

Another driver for the observed concentration pattern might be leaf surface wetness. Peatlands in general, particularly during colder parts of the year, are moist environments where ammonia can easily be taken up by wet surfaces. On the other hand, it is released back to the atmosphere when surface water, e.g. dew, evaporates during morning and midday hours (Walker et al., 2006; Wentworth et al., 2014), which then causes again rising ammonia concentrations as observed in this study (Fig. 4).





Other authors observed the concentration peak earlier in the morning, e.g., Walker et al. (2006) and Wolff et al. (2010) at arable land and grassland, respectively. The observed peak in our study might have been shifted because of the much higher humidity at our peatland site, indicated by small water pools, causing a longer duration of the evaporation process.

The monthly integrated ammonia concentration of 16.8 ppb in March from QCL measurements was in good agreement with

those values measured by DELTA denuder and passive samplers. The latter approaches resulted in 14.5 and 15.2 ppb for DELTA and passive samplers, respectively, indicating their robustness and validity as low-cost methodologies for longterm air quality monitoring. As the time of exposure of DELTA denuders and passive samplers was not consistent with our QCL measurements due to instrument failures and campaign duration in February, April, and May, as well as due to highly variable concentrations during that time, we could not directly compare QCL numbers with those from the monthly

integrating methods.

## 3.2 Ammonia exchange and its biophysical controls

Half-hourly measured ammonia fluxes ranged mainly within −80 and 20 ng N m$^{-2}$ s$^{-1}$ with only very few values as low as −300 and as high as 300 ng N m$^{-2}$ s$^{-1}$ (upward fluxes positive, Fig. 3, lower panel). On average the peatland was a sink with a mean flux of −17.4 ng N m$^{-2}$ s$^{-1}$. At the beginning of the campaign (Period I and II), ammonia deposition was consistently

recorded (<−20 ng N m$^{-2}$ s$^{-1}$; Fig. 3), while in Periods III and IV the average deposition decreased to values >−10 ng N m$^{-2}$ s$^{-1}$ and the exchange became clearly bi-directional.

We observed considerable diurnal variability in ammonia fluxes throughout the campaign. The average diurnal flux showed moderate uptake around −25 ng N m$^{-2}$ s$^{-1}$ from midnight to 10 a.m., near-neutral exchange around noon, and highest uptake of −40 ng N m$^{-2}$ s$^{-1}$ in the late afternoon and early evening hours (Fig. 6 and 7). Separated by episodes, Period II showed the

largest amplitude, whereas Period IV revealed only little variability over the daily course. Separated by surface wetness, considerably higher uptake was observed in the late afternoon and early evening when there was no precipitation recorded at the site than during times when it was raining. Furthermore, ammonia exchange shifted from around zero to net emissions at noon during rain events. Typically rain events were mostly associated with winds from the south-west, where the influence of the agriculture is lowest. We also used the elapsed time after the last recorded rain as a proxy for leaf surface wetness

(Fig. 6). We found that higher ammonia uptake coincided with a larger number of days passed since last rain.

Springtime ammonia uptake at sites that were highly influenced by fertilization and other local sources of ammonia in the surrounding area has been reported earlier, e.g. by Mosquera et al. (2001), who found considerable average deposition fluxes at their semi-natural grassland site. Beside management, they showed that higher surface wetness in spring amplified local ammonia deposition, while net emission was typically found only in summer (see also Wichink Kruit et al., 2007). An

undisturbed peatland site is likely to be a higher ammonia sink than managed grasslands due to a lower nitrogen status and therefore lower ammonia compensation point. However, with the chronically high atmospheric nitrogen loads caused by agriculture over several decades our peatland is presumably not such an efficient sink anymore. With this assumption and



the decreasing uptake over the progression of the measurement campaign (box plots in Fig. 3), saturation effects might have played a role in biosphere-atmosphere exchange characteristics. Flechard and Fowler (1998) already showed that peatland vegetation under its common wet conditions may not necessarily be an 'almost perfect sink' as was reported by Duyzer (1994) due to nitrogen saturation effects in heathland plants caused by persistently high ammonia deposition induced by local sources.

Regarding the diurnal flux patterns observed in this study, significantly different exchange characteristics have been reported elsewhere. For example Horvath et al. (2005) and Wichink Kruit et al. (2007) found highest deposition rates in early morning hours due to dew formation at their grassland sites with decreasing deposition and even emission afterwards through drying leaves and stomatal release. Their finding, i.e. the ecosystem emits only under dry conditions is contrary to our observations (Fig. 6 and 7).

Regarding the entire campaign, a significant tipping point in ammonia exchange was found on 15 March, where higher deposition rates changed to much lower deposition and bi-directional exchange (*cf.* Fig. 8 and 11; Section 3.3). This tipping point followed 10 days after the onset of spring, occurring on 5 March, as indicated by $CO_2$ uptake shown as significant gross primary productivity (GPP) in Hurkuck et al. (2016) and was mainly triggered by a significant weather change. A huge pressure drop and daily minimum air temperatures exceeding 5 °C for the first time in that particular year were recorded. While conditions were dry until 15 March, 25 mm precipitation was recorded during the following nine days. Thus, a likely reason for decreasing uptake after 15 March might have been a shift of the stomatal compensation point of the local active peatland vegetation due to higher temperatures as was shown by Milford (2004). Similar to findings in Kim et al. (2011), precipitation after 15 March significantly reduced ammonia concentrations in ambient air while the ratio of wet to dry deposition considerably increased.

The tipping point from higher to much lower ecosystem ammonia uptake was also observed in the course of deposition velocities ($v_d$) (Fig. 8). Mean $v_d$ is defined as the ammonia flux divided by concentration over a given period and is still a valuable indicator to verify dry deposition inferential models such as AUSTAL2000 (Janicke, 2002), even in the case of bi-directional exchange (in the case of emission flux, $v_d$ is negative). In our study mean $v_d$ decreased from approx. 0.5 cm s$^{-1}$ at the beginning of the campaign to approx. 0.2 cm s$^{-1}$ on 15 March and remained relatively constant until the end of the campaign. These values are significantly lower than those reported by Schrader and Brümmer (2014), who found a median and weighted average of 0.7 and 0.9 cm s$^{-1}$, respectively, for the category 'semi-natural' ($n$=19) in their literature review. In that study, single site values of $v_d$ were ranging between 0.1 and 1.8 cm s$^{-1}$.

The observed tipping point on March 15 is also likely another indicator of increased canopy resistance ($R_c$) over time, which effectively reflects the increase of the canopy compensation point, which may be related to both stomatal and non-stomatal influences. Over a coniferous forest, Wyers and Erisman (1998) described an increase in nocturnal $R_c$, which was interpreted as a consequence of pre-deposited ammonia leading to an alkaline saturation of leaf surfaces. For drying canopy, the same authors also found an increase in $R_c$ in a heathland study (Erisman and Wyers (1993), which is consistent with Walker et al.





(2006). Our observations showed that the tipping point occurred in a relatively dry period and also that ammonia uptake remained low during rain events ten days later. This might indicate that at a certain point ammonia exchange was partly no longer controlled by surface wetness but by the pH on non-stomatal surfaces. However, stomatal plant physiological effects in the form of the seasonal onset of $CO_2$ uptake as mentioned above was likely be caused by the change in weather and

therefore be probably the main reason for the tipping point.

The statistical significance of the ammonia flux dependency on meteorological variables when classified into different ranges of values was checked by means of a Kruskal-Wallis-Test (Tab. 2). The null hypothesis of identical population was rejected in all cases when the p-value was below the commonly used significance level of $\alpha = 0.05$. A Post-hoc-test confirmed that the distributions in all groups, except for '1−2 days' and '2−5 days' in the category 'days after last rain', were

significantly different. Thus, all criteria, i.e. the biophysical factors air temperature, precipitation, surface wetness, and radiation, had a statistically significant influence on ammonia exchange. However, fluxes were not well correlated with air temperature, radiation or concentration ($R^2 < 0.1$) when using simple regression analysis. Milford et al. (2001) reported that these variables usually regulate the ammonia exchange but could not find a good correlation between flux and temperature or radiation either because of many non-linear interrelations (*cf.* Milford, 2004; Yamulki et al., 1996). This can be confirmed by

visually inspecting the diurnal cycles shown in Fig. 7. The course of ammonia fluxes is more closely coupled to net radiation and $u_*$, whereas peaks in concentration, although rather forming a bimodal pattern, follow the shape of air temperature with a 2−3 hour lag. Furthermore, solubility of ammonia is related to temperature, which in turn drives the opening of stomata (Fowler et al., 1998), although the response of radiation is much stronger and the temperature effect on stomatal conductance is often confounded by the stronger vapor pressure deficit effects. With increasing temperature ammonia is less dissociated

in the available water reservoirs and the plants are able to easier release it to the atmosphere.

Radiation and temperature are also driving local turbulence, i.e. $u_*$, which appears also to be a controlling factor for both ammonia concentration and fluxes (Fig. 9). While concentrations were lowest under low turbulence, even when analyzed for single periods, maxima were observed in the 0.1 to 0.2 m s$^{-1}$ class with slightly decreasing values under increasing turbulence, which is a natural phenomenon due to better mixing and transport in the boundary layer, whereas low

concentrations under low turbulence at night are likely an artefact of the local fertilizer management that is usually applied during the day. Highest uptake fluxes, however, were found under low $u_*$, whereas lowest uptake was found under high $u_*$. This is confirmed by separating day and nighttime data with highest uptake of ammonia clearly occurring during the night (left and middle panel of Fig. 9). The same patterns were found when plotting $u_*$ vs. normalized fluxes, e.g. $v_d$ (not shown). Decreasing uptake with increasing $u_*$ is rather an unusual finding and could be indicating that the control of turbulence is

significantly masked by the strong influence of plant physiological regulation (VPD response of stomata) and agricultural practices at our study site with co-occurring gradual ammonia saturation of the leaf surface. Further, at low turbulence local transport from nearby fields and farms into the peatland occurs, at high turbulence a high proportion is going directly from the sources into long-distance transport. At daytime this is mirrored by the flux, thus it seems like the influence of $u_*$ on the



ammonia flux is small compared to concentration. At nighttime there is no tendency except a generally higher flux compared to daytime, which may be an effect of wet surfaces rather than any turbulence effect. It has been reported – even for agricultural sites – that medium to high turbulence favors the magnitude of exchange fluxes, regardless of direction, i.e. emission or deposition (*cf.* Brümmer et al., 2013).

The interdependency of the ammonia flux and concentrations is shown in Fig. 10. We separated emission from deposition periods and bin-averaged the concentration data. Our observations are consistent with Milford (2004) who also found both increasing emission and deposition under elevated concentration, also when separating by wind direction (data not shown). It remains a matter of speculation whether the flux controls the concentration or vice versa as the relationship is highly controlled by plant nitrogen status and at least to some extent by biometeorological variables as mentioned above. However,

Milford's (2004) statement that the concentration may still determine the flux during deposition periods, whereas during emission periods it may be the ammonia flux itself, which is controlling the concentration, may also be applicable for our site. However, more likely is a coincidence of flux drivers and high concentration levels which were high due to advection from the local sources. What is needed at this point is not just an observation on one field and one ecosystem, but a landscape-scale or regional-scale model of emission, dispersion, chemistry, exchange and deposition, which makes it

possible to work on this question interactively.

### 3.3 Measured vs. modeled fluxes and cumulative exchange

The comparison of measured and modeled daily mean and cumulative half-hourly ammonia fluxes is given in Fig. 10 and 11. We found a considerable mismatch between modeled and measured fluxes with the latter showing higher uptake in Period I and lower uptake in Period III than model outputs. In contrast, during Periods II and IV measured and simulated

fluxes run fairly parallel, with the exception of a short period of overestimated deposition during last week of March, indicating that, on average, during these times the model is able to reproduce the measured fluxes well. The interdependency of modeled ammonia fluxes and measured concentrations is very similar to the measured ones except that the model does not exhibit any emission and the fluxes are generally lower with increasing concentrations (Fig. 10). We do not see the larger measured deposition fluxes during Periods I and II as being indicative of faulty measurements. Instead, we suspect that under

the local pollution climate during the measurement period, the model predicts a too large non-stomatal resistance ($R_w$). The Massad et al. (2010) parameterization uses a so-called acid to ammonia ratio (AAR), i.e. the molar ratio of the sum of $HNO_3$, $SO_2$, and $HCl$ concentrations to the $NH_3$ concentration, to scale the minimum allowed $R_w$ in the model. The very low AAR (0.07 to 0.11) measured at the peatland site increases the baseline (acid ratio = 1) minimum $R_w$ of 31.5 s m$^{-1}$ roughly 9- to 14-fold. A comparison with non-stomatal resistances inferred from night-time measured fluxes and modeled $R_a$ and $R_b$ (*cf.*

Wichink Kruit et al., 2010) reveals that this is too large for periods I and II (not shown here); however, in Periods III and IV, when temperatures rise and thermodynamic equilibria are shifted towards ammonia in the gas phase – increasing apparent $R_w$ and decreasing deposition on the external leaf surfaces – the model is able to reproduce the measured $R_w$ well. Furthermore,





$HNO_3$, $SO_2$ and HCl concentrations were only measured on a monthly basis using DELTA denuders, thus introducing some uncertainty to the estimated acid ratio and therefore modeled $R_w$.

The cumulative exchange after approx. 9 weeks of observation resulted in total deposition of 911 and 857 g $NH_3$-N $ha^{-1}$ for measurements and model output, respectively, thus matching relatively well over a longer period despite considerable

deviation in Periods I and II. Hurkuck et al. (2014) estimated annual ammonia deposition of 8.5 kg N $ha^{-1}$ at the same study site using denuder filter systems in combination with inferential modeling. They found considerable seasonal variability and also accounted agricultural practices in the surrounding area as the main driver for ammonia exchange variability. Extrapolating our measurement-based campaign total time proportionally to an entire year results in a net deposition estimate of approx. 5.3 kg $NH_3$-N $ha^{-1}$. Reasons for the mismatch might be the fact that our measurements are based on a much

higher time resolution leading to a more accurate representation of emission components in the net signal of the eddy-covariance fluxes. Additionally, $R_w$ is based on a much more complex parameterization in Massad et al. (2010), including AAR and a correction for temperature and leaf area index, than those methods used in Hurkuck et al. (2014). However, even with an approx. 3.2 kg N $ha^{-1}$ $yr^{-1}$ lower ammonia depostition, our study site remains a substantial total nitrogen sink. Adding the numbers presented in Hurkuck et al. (2014), i.e. another 2.4 kg N $ha^{-1}$ $yr^{-1}$ dry deposition of HONO, $HNO_3$, and

$NH_4NO_3$ aerosols as well as 14 kg N $ha^{-1}$ $yr^{-1}$ as wet deposition, total nitrogen deposition results in approx. 21.7 kg N $ha^{-1}$ $yr^{-1}$, thereby more than four times exceeding the ecosystem-specific critical load of 5 kg N $ha^{-1}$ $yr^{-1}$ (*cf.* Bobbink et al., 2010; UNECE, 2004).

## 4 Conclusions

Eddy-covariance flux measurements of ammonia using a QCL in combination with an 'inertial inlet box' were conducted at

a peatland site in an agricultural landscape. This methodology has high potential for (1) an establishment in long-term observation networks with the aim to improve nitrogen budgets and transfer calculations at local and regional scale as well as (2) providing deeper insight into the mechanisms of ammonia transfer and the ecosystems' responses to ammonia loads in the atmosphere by offering continuous flux observations at unprecedentedly high temporal resolution. In the present study, we interpret changing diurnal patterns of ammonia concentration and fluxes as well as a tipping point followed by decreasing

deposition velocities and increasing canopy resistance, as a sign of non-stomatal leaf surface $NH_3$ saturation in response to elevated $NH_3$ from agricultural activities (i.e. manure spreading), but also by delayed plant physiological effects after the onset of the growing season. Temperature, radiation, and surface wetness were identified to partially regulate ammonia exchange at the site; however, the seasonal concentration pattern was clearly dominated by emissions from agricultural practices in the surrounding area. Overestimations of non-stomatal resistances due to low acid to ammonia ratios were

assumed to be responsible for deviations between modeled and measured flux estimates. On a total cumulative campaign basis the estimated ne $NH_3$ exchange differed only by 6 % between the model and independent flux measurements. Lower QCL-based deposition values than those of using denuder systems were likely due to a better representation of the emission



component in the net signal of eddy-covariance fluxes as well as better adapted site-specific model parameters, particularly $R_w$. Further research is needed on long-term stability of the QCL system and on avoidance of unintended reactions of ammonia within the inlet and sample tube.

**Acknowledgements**

5   Funding for this study from the German Federal Ministry of Education and Research (BMBF) within the framework of the junior research group NITROSPHERE under support code FKZ 01LN1308A is greatly acknowledged. We thank Jeremy Smith, Jean-Pierre Delorme, Ute Tambor, Andrea Niemeyer and Dr. Daniel Ziehe for excellent technical support and conducting laboratory analyses of denuder and filter samples, respectively.



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





**Tables**

**Table 1: Characterisation for four subperiods of the measurement campaign (I to IV) with differenet NH3 concentration and weather regimes.**

| Period | Time | $\bar{c}_{NH_3}$ (ppb) | SD of $c_{NH_3}$ (ppb) | Minimum $c_{NH_3}$ (ppb) | Maximum $c_{NH_3}$ (ppb) | $\overline{T_a}$ (°C) | P (mm) | $R_n$ (MJ m$^{-2}$ d$^{-1}$) |
|---|---|---|---|---|---|---|---|---|
| I | Feb 18 – Mar 04 | 11.3 | 5.6 | 1.2 | 34.7 | 4.5 | 6.4 | 1.71 |
| II | Mar 05 – Mar 15 | 22.2 | 12.1 | 3.4 | 87.8 | 6.3 | 3.7 | 4.86 |
| III | Mar 16 – Apr 10 | 16.5 | 11.7 | 2.2 | 87.4 | 8.6 | 35.5 | 5.19 |
| IV | Apr 29 – May 07 | 8.0 | 4.2 | 1.7 | 21.0 | 10.3 | 15.5 | 8.14 |



**Table 2: Data classification and results of Kruskal-Wallis test (see Section 3.2 for details).**

| Meteorological variable | Groups | | | p-value | Post-Hoc |
|---|---|---|---|---|---|
| Air temperature | <5°C | 5 – 10°C | >10°C | <0.001 | All differ |
| Precipitation | 0 mm | >0 mm | | <0.001 | All differ |
| Days after last rain | 1-2 d | 2 – 5 d | >5 d | <0.001 | 1=2 |
| Net radiation | <0 W m$^{-2}$ | 0 – 150 W m$^{-2}$ | >150 W m$^{-2}$ | <0.001 | All differ |



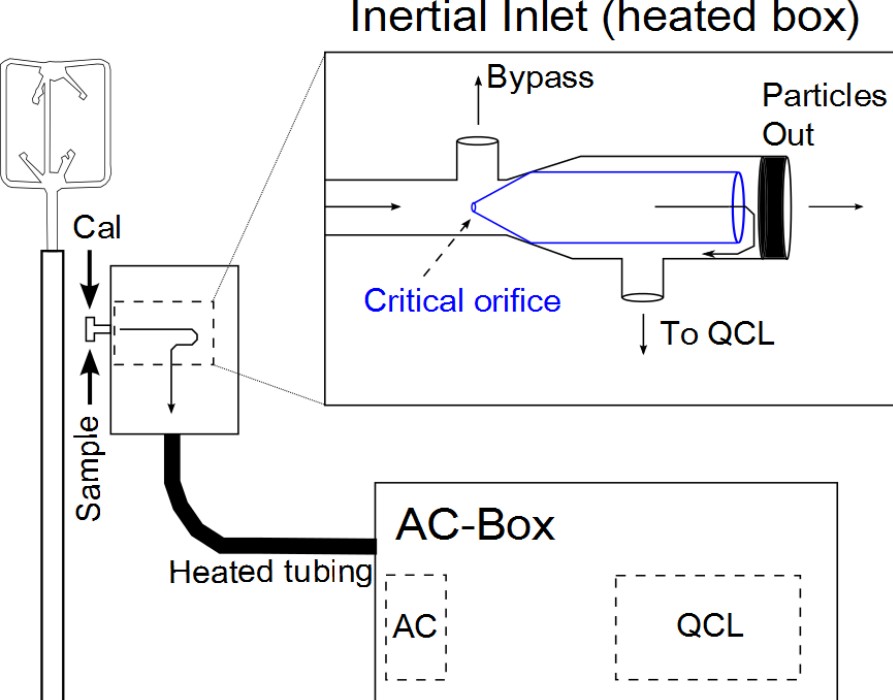

**Figure 1: Schematic overview of the measurement setup. An ultrasonic anemometer is mounted closely to the heated 'inertial inlet' box containing a critical glass orifice to reduce the pressure regime inside the sample line. After passing the critical orifice, a sharp turn of the sample line leads to a reduction of particles (>300 nm) of approx. 50 %, thereby reducing unintentional chemical interactions. The heated tubing leads the sample air to the QCL, which is housed in an air-conditioned box.**





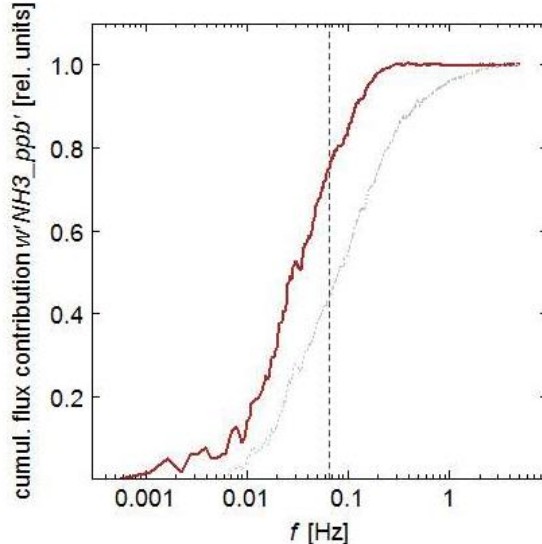

**Figure 2: Normalized flux ogives, i.e. cumulative cospectra, of sensible heat (grey) and NH$_3$ flux (red). Data were recorded on 18 February 2014, 03:30 – 04:00 pm. The vertical dotted line indicates the frequency at which the two ogives are compared to determine the empirical high-frequency damping factor for the NH$_3$ flux.**





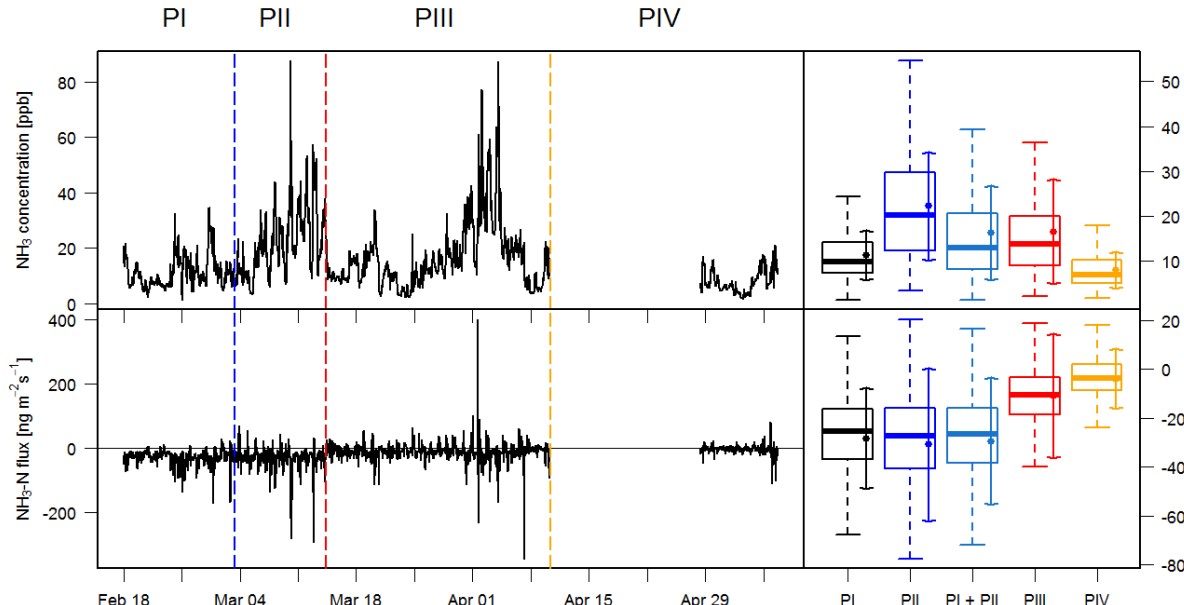

**Figure 3: Half-hourly measured ammonia concentrations and fluxes. Vertical lines indicate beginning of periods listed in Table 1. Each period is represented by a boxplot with bold horizontal lines showing the median, fine horizontal lines indicating lower and upper quartile values, whiskers representing 1.5 times the interquartile range and dots with arrows indicating the mean and standard deviation.**





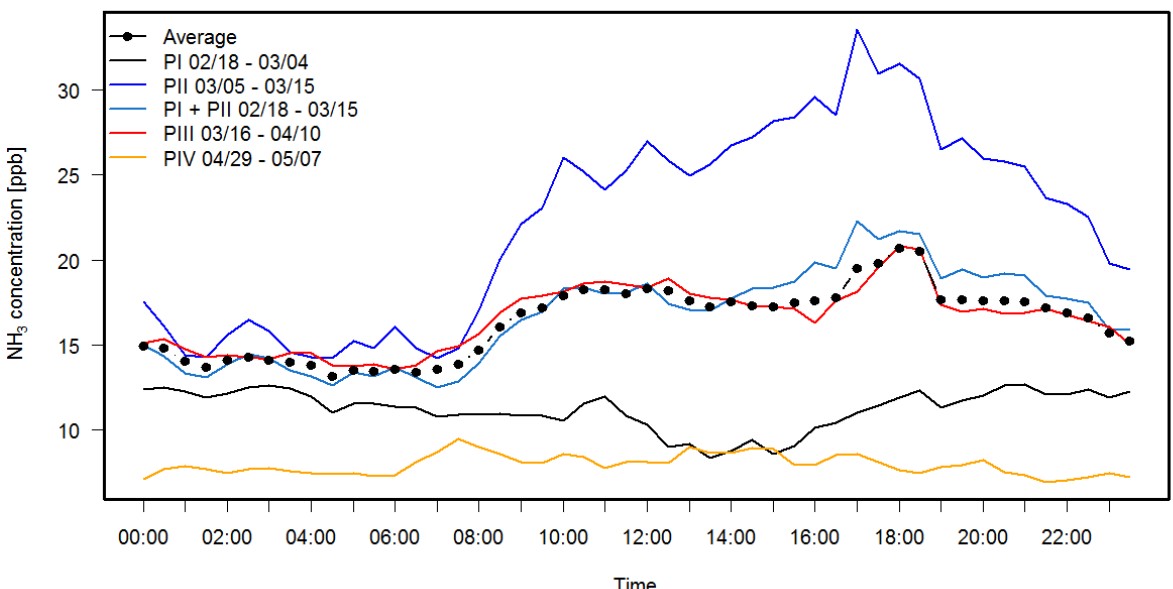

**Figure 4: Mean diurnal variation of ammonia concentrations separated into different periods.**





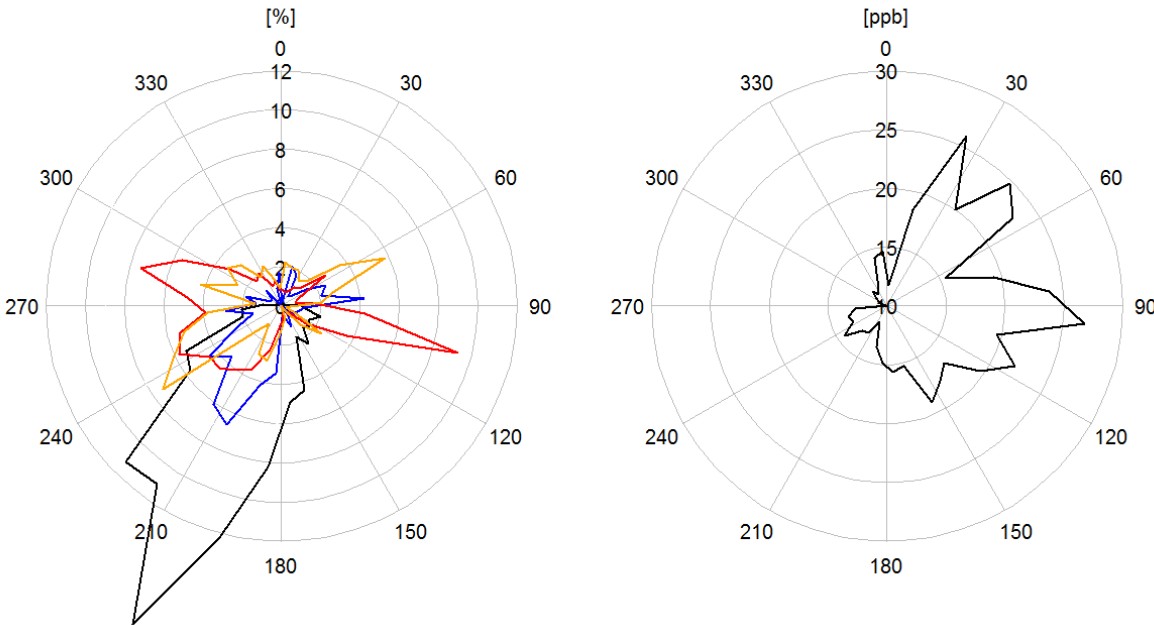

**Figure 5: Frequency distribution of wind direction for each period (left panel, PI (02/18 – 03/04) black, PII (03/05 – 03/15) blue, PIII (03/16 – 04/10) red, PIV (04/29 – 05/07) orange) and ammonia concentration (right panel) for each 10° wind sector.**





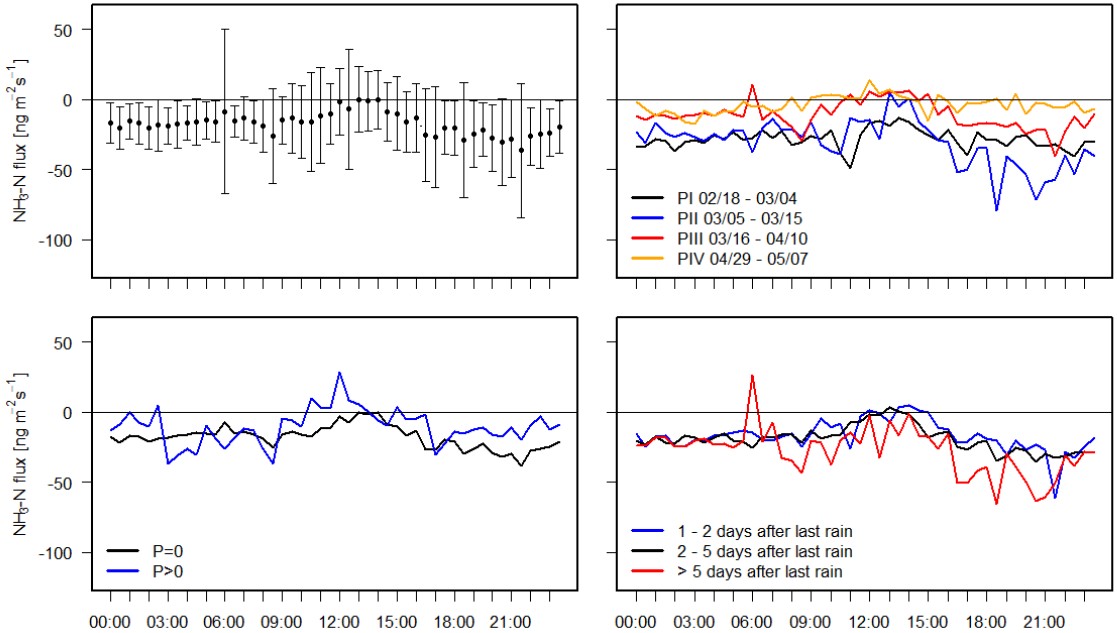

**Figure 6: Mean diurnal variation of ammonia fluxes with standard deviation (upper left panel), separated by periods (upper right panel), by precipitation (lower left panel), and by days after last rain (lower right panel).**





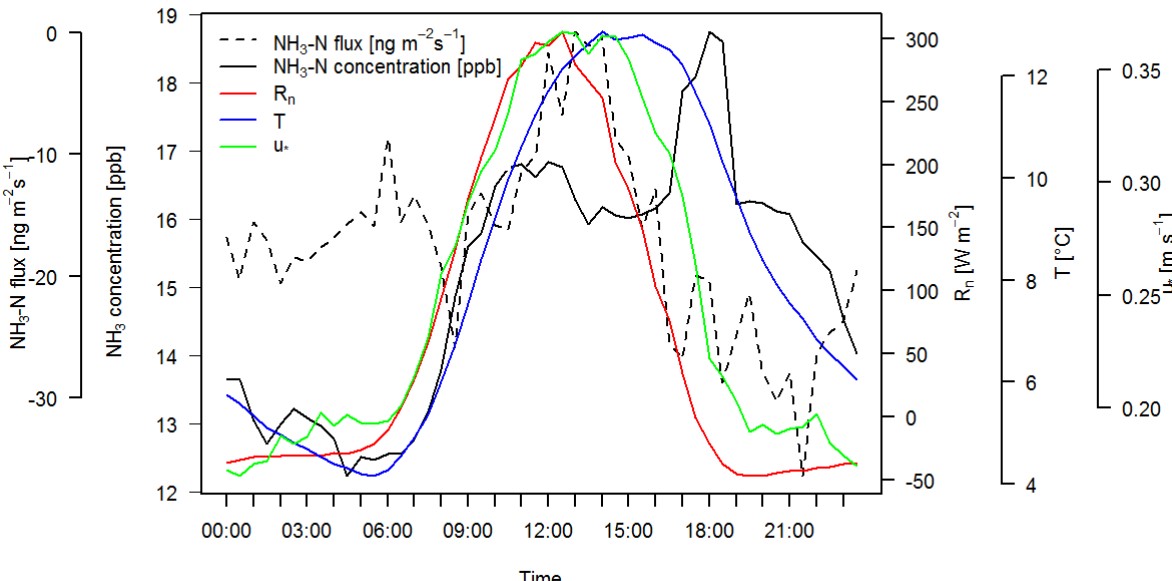

**Figure 7: Mean diurnal cycles of ammonia concentration, ammonia flux, net radiation, air temperature, and friction velocity.**





**Figure 8: Half-hourly NH₃ deposition velocities expressed as NH₃ flux divided by NH₃ concentration during the measurement campaign. Vertical lines indicate the start of Periods II, III, and IV as listed in Table 1.**



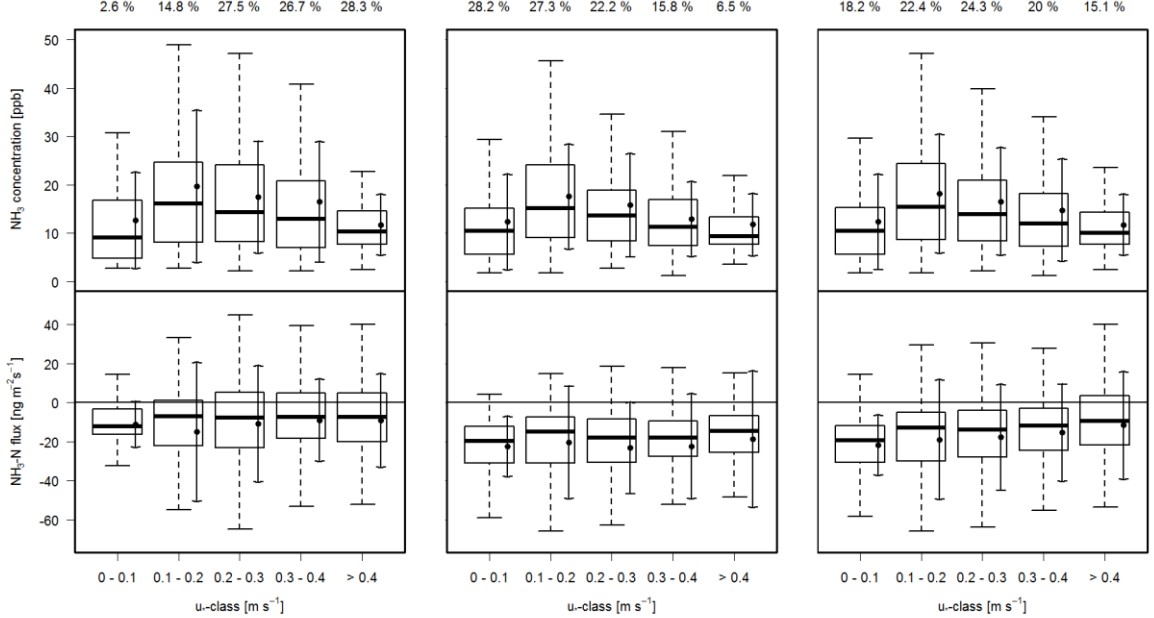

**Figure 9: Dependency of ammonia concentrations and fluxes on $u_*$ with panels from left to right showing daytime ($Rn > 20$ W m$^{-2}$), nighttime ($Rn < 20$ W m$^{-2}$), and all data, respectively. Bold horizontal lines in boxplots show the median, fine horizontal lines indicate lower and upper quartile values, whiskers represent 1.5 times the interquartile range and dots with arrows indicate the mean and standard deviation. Values in upper row specify percentages of data in the respective $u_*$ category given below.**





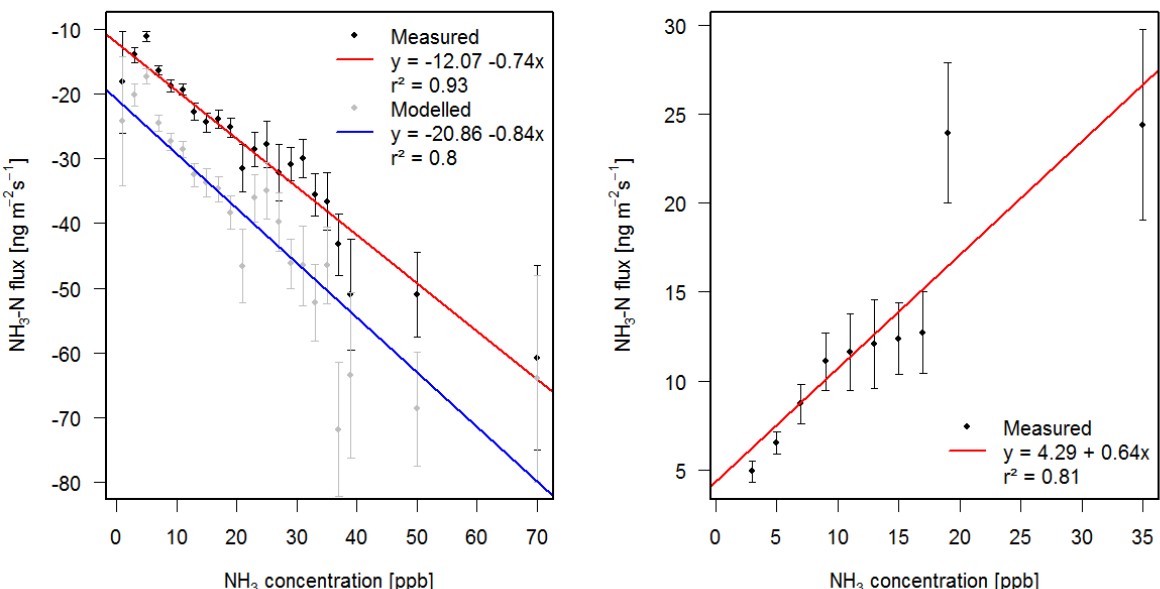

**Figure 10: Dependency of NH₃ fluxes, measured and modelled, on bin-averaged NH₃ concentrations (bin sizes are 2 ppb for**
**concentrations < 40 ppb and < 20 ppb, in the left and right panel, respectively; for concentrations > 40 ppb bin sizes are 20 ppb)**
**during deposition (100% of the modelled and 84 % of the measured data, left panel) and emission periods (16% of the measured**
**data, right panel), error bars indicate standard error.**





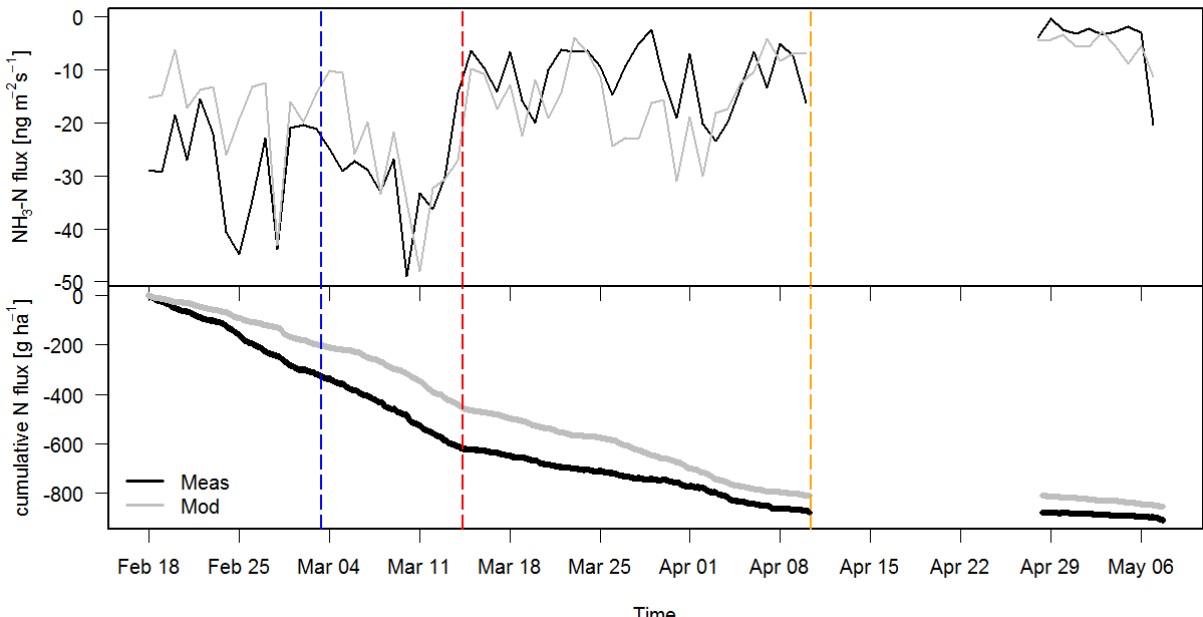

**Figure 11: Comparison of measured and modeled daily mean NH₃ fluxes (upper panel) and cumulative NH₃ flux (lower panel) based on half-hourly data during the measurement campaign. Vertical lines indicate the start of Periods II, III, and IV as listed in Table 1.**