# Peer review of "Surface-atmosphere exchange of ammonia over peatland using QCL-based eddy covariance measurements and inferential modeling"

_Atmospheric Chemistry and Physics, 2016_

## Referee Comment (RC2)

**« Surface-atmosphere exchange of ammonia over peatland using QCLbased eddy covariance measurements and inferential modelling »**

**General comments**

This manuscript presents the results of a 2.5 month long ammonia flux measurement campaign over a peatland with an eddy covariance method implementing a new inlet system intended to reduce aerosol and henceforth water interactions with ammonia on tube walls, a limiting issue for flux ammonia measurements by eddy covariance with quantum cascade laser until now. The authors show quite convincingly that the new inlet system is adapted for  $NH_3$  flux measurements by Eddy Covariance but recognise a potential bias due to evaporation of  $NH_4^+$  from small aerosols in the heated line which they estimated to be lower than 21% on the concentration but did not give an estimation on the flux uncertainty. The dynamics of the measured ammonia concentration and fluxes are analysed in terms of correlation with meteorological conditions and discussed with regards to the influence of nearby farms and ecosystem functioning. They investigate the relationship between  $NH_3$  concentrations and fluxes but without using the useful framework of resistance modelling, which they use only for estimating the cumulated ammonia deposition.

This manuscript reports thoroughly designed and conducted experiments and is to my knowledge a unique reporting of continuous  $NH_3$  flux measurements by eddy covariance over such a long period. The analysis and discussion of the flux and concentration dynamics is interesting but lacks overall quantification of the emission potential and surface resistances. Moreover, the analysis would be greatly improved by better discussing the flux uncertainties and analysing the hourly dynamics of the deposition velocity rather than the flux which is concentration dependent. In my opinion, this manuscript could be greatly improved by answering the following issues, prior to publication:

- The quality of the flux measurements which is critical due to the novelty of the inlet system is however difficult to figure out completely for the reader. The way the time lag is calculated is not completely clear in the current manuscript, and the lag is an essential parameter in the flux which could change its magnitude by a large fraction, especially with noisy signals. It would be good to show some covariance peaks and may be the dynamics of the lag (in a supplementary material section?). Similarly, a median value of the high frequency damping factor is given without much details, nor discussion and comparison to previous literature (Ferrara et al., 2012;Whitehead et al., 2008). I would suggest showing the dynamics of the high frequency damping (it could be a box plot of hourly values for instance). The random uncertainty is also given as a 15% estimate, but there are some methods to evaluate the uncertainty in the flux, which are especially designed for fluxes with large instrumental noise. I would suggest to get example on Langford (2015), and to report flux detection limit. Some of these methods can be turned on in EddyPro, so this should not represent too much work.
- The analysis of the correlation of the fluxes with the meteorological conditions is quite instructive but lacks a deeper insight into the surface exchange parameters. Indeed, a first essential test is a comparison of the deposition velocity  $V_d(z)$  with the maximum deposition velocity for ammonia  $V_{dmax}(z)$ , which represents the maximum transfer rate and can simply be evaluated as  $(R_a(z) + R_b \{NH_3\})^{-1}$ , where  $R_a(z)$  and  $R_b \{NH_3\}$  are the aerodynamic and boundary layer resistances for ammonia, respectively (e.g. (Loubet et al., 2012)). Similarly, analysing the statistics of the deposition velocity would probably give more insight into the exchange processes than the ammonia flux because of the large variability of the atmospheric concentration which is influenced by the local sources. An analysis of the daily variations of the deposition velocity would be very instructive. This would especially be helpful for understanding the links between Figure 9 and 10, which is not clear in the current manuscript.
- The resistance analogy would also be very helpful to better evaluate the surface emission potential  $\Gamma(z_0)$ and its dynamics. Indeed the canopy compensation point could be estimated as  $C_c = F_{\text{NH3}} \times (R_a(z) + R_b\{\text{NH}_3\}) + C_{\text{NH3}}(z)$ , and the emission potential retrieved from that using the thermodynamical gas-to-liquid and acid-base equilibrium constants. See Sutton *et al.* (2009), Loubet *et al.* (2012) or Personne *et al.* (2015). The compensation point could be also estimated by analysing daily flux versus concentration relationships (similar to Figure 10 but for each day). This would ease a lot the understanding of the seasonal evolution of the ammonia flux, and its relationship with ecosystem functioning. It will as well help controlling the quality of the flux measurements.
- In a complementary analysis, surface conductance  $(g_c, the inverse of the resistance R_c)$  could be estimated (assuming a zero emission potential in the canopy under deposition conditions) based on

 $V_{\rm d}(z)$  and  $V_{\rm dmax}(z)$ . Indeed, then  $g_{\rm c}^{-1} = V_{\rm d}(z)^{-1} - V_{\rm dmax}(z)^{-1}$  (Massad et al., 2010). This would withdraw part of the influence of  $u_*$  on the exchange dynamics, which is embedded in  $V_{\rm d}(z)$ , and hence better show changes in ecosystem exchange parameters, and especially cuticular exchange.

- Figure 10 puzzles me for several reasons, but I might not have understood correctly how it was built:
  - (1) I cannot figure out why the modelled flux is smaller than the measured one with a constant offset (in Fig 10-right), while it shows larger values in Figure 11 at the beginning. I also interpret a constant offset as an additional pathway with a constant flux, but cannot reconcile this with the model of Massad *et al.* (2010) as used here.
  - (2) I cannot understand why no intercept (flux crossing the zero line) can be seen in Figure 10(left) while in Fig 8 we see negative  $V_d(z)$ . It is probably due to the separation between emission (Fig10. Right) and deposition (Fig10. Left) fluxes. The authors should clarify how Fig. 10 is constructed. Especially important would be to show some dynamics of the daily flux and concentration with emissions. Currently only averages are shown (except for Vd(z)) and no emissions can be seen except in Fig. 6c and in the error bars of Fig 6a. An example of daily dynamics would be most helpful.
- The authors should also discuss further, based on more quantified surface parameters, whether the flux is linked to a surface compensation point or some other features. Especially, the magnitude of the advection fluxes should be evaluated. Indeed, if the NH3 concentration peaks of up to 85 ppb are due to concentration advected from nearby farms and agricultural activities, advection fluxes can be expected to be large. To evaluate these a footprint model could be used, the advection fluxes would then be the footprint of the farm (or fields spread with organic manure) multiplied by the source strength of these, which could be evaluated by a simple emission factor analysis. This would allow evaluating whether advection is an issue or not. I would suggest the authors to look at (Hensen et al., 2009;Loubet et al., 2009).

**Detailed comments**

- Section 2.1: The ground pH and NH4+ are important parameters for interpreting NH3 fluxes. Were any of these measured? If so, they should be reported.
- P3L29: The authors should rather use "mixing ratio" rather than concentration. Also are these expressed per mol of dry air or per mol of ambient air? Please discuss this point as this makes a difference in the flux calculation which should be done with mixing ratio per mol of dry air (Gu et al., 2012;Kowalski and Serrano-Ortiz, 2007). Especially important is the dilution effect due to water.
- P4L30. What is the inlet box size? Could you discuss briefly the potential impact on the flux measurements?
- P5L7-12: The way the two time-series were shift is not sufficiently detailed here. Especially, could the authors explain how the expected time lag was chosen? Also, it would be important to show that this procedure did not strongly affect the flux. Could the authors please discuss this point further? The authors may consider adding a graph showing several covariance peaks for emission and deposition conditions.
- P5L15-20. Please give more details on the damping factor and how it evolved during the campaign. Some additional graphs could be proposed in a supplementary material section.
- P5L20-23. The uncertainty on the flux is critical for NH3 which is not a routine measurement. I suggest taking example on (Langford et al., 2015) and related references for computing the error on the flux and evaluating the flux detection limit.
- P5L29: Clarify if gap-filling was also performed for NH3 and if so how.
- P6L6: Be careful that  $R_a$  is a function of the measurement height z. Consider using the notation  $R_a(z)$ .
- P6L7-12: The parameters of the Wesely model should be given here: the minimum resistance and the response to radiation.
- Section 3.1: Since the local farms and agricultural fields play an important in the interpretation of the mixing ratios.
- P7L27-28: This sentence is unclear. Please rephrase.
- P7L33-34: The work of Flechard *et al.* (1999), Wu *et al.* (2009), and Burkhardt et al. (2009) should be mentioned here.
- P9L1-5. Could you be more quantitative here? Are the levels comparable with Duyzer (1994)? What is the amount of NH3 received in this study? What would be the ecosystem compensation point predicted by Massad *et al.* (2010) with this deposition? Please also discuss this issue with reference to Wu et al. and Burkhardt et al.

- P9L12: It is difficult to see a change on Fig. 8. Please consider re-graphing this figure.
- P9L17: A shift of the stomatal compensation point could be evaluated by retrieving the daily compensation point as explained in the general comments. Two methods are possible:
- P9L19-20: The data on wet to dry deposition are not shown here. Please consider adding these to the supplementary material or at least giving numbers to support the sentence.
- P9L21-33 and P10L1-5: Showing the maximal exchange velocity  $V_{dmax}(z) = (R_a(z) + R_b \{NH_3\})^{-1}$  would be important to show the plausibility of the flux. Moreover, you can then calculate the canopy resistance  $R_c$  or the canopy conductance  $g_c$  as  $g_c^{-1} = V_d(z)^{-1} - V_{dmax}(z)^{-1}$ , during deposition periods (especially at the start of the campaign). This would probably better show the dynamics of the ecosystem exchange parameters, together the surface emission potential  $\Gamma(z_0)$  which could be estimated from the canopy compensation point  $C_c(z_0) = F_{NH3} \times (R_a(z) + R_b \{NH_3\}) + C_{NH3}(z)$ , with the relationship  $C_c(z_0) = \Gamma(z_0) \times 10^{-3.4362 + 0.0508 T(z_0 \text{ in }^{-C})}$ . This will probably help understanding the surface exchange dynamics and also test the plausibility of the flux and concentration measurement as  $C_c$  should remain positive.
- P10: This study on the parameters influencing the ammonia exchange would benefit from being made on the deposition velocity which would less depend on the variable atmospheric concentration.
- P10L7: explicit the term  $\alpha$ .
- P10L28: Vd would be indeed good to show together with the flux!
- P10L29-33: The daily evolution of the flux with decreasing deposition or even small emission around noon and deposition at night could be a consequence of a stomatal or ground compensation point which evolves following the daily surface temperature  $(T(z_0))$  pattern and is much larger at noon than during the night. This explanation also reconciles the observed dependency of the flux to  $u_*$ , observed both during day and night (Fig. 9): indeed, the surface temperature  $T(z_0)$  will increase with increasing  $u_*$  at night with clear sky due to better mixing and hence less radiative cooling. During the day, the increase of surface temperature is mostly linked with incoming solar radiation and peaks at the same time as  $u_*$ .
- P11L2-4: Indeed.  $V_{\text{max}}(z)$  is a measure of this exchange velocity and comparing  $V_d(z)$  with  $V_{\text{max}}(z)$  would easy this discussion.
- P11L5-15; I cannot figure out how to interpret this Figure. I would suggest the authors to try showing the same relationship without separating emissions and depositions to show whether there is or not a compensation point and try to evaluate it. It would be interesting to try to build a figure for some representative days based on 30-min data and see whether the change from deposition to emission appears at a given concentration (the definition of the compensation point).
- I can imagine that the advection indeed could indeed lead to such dependency. But it would then be interesting to evaluate the potential for advection based on a simple footprint model such as Kormann and Meixner (2001), which is available as an excel spreadsheet (Neftel et al., 2008). If you just multiply the value of footprint of the surrounding agricultural field or farms by an estimated magnitude of the NH3 fluxes there, you could evaluate the potential effect of the advection on your flux. See also in Loubet *et al.* (2009) and in Sutton *et al.* (2009) for a discussion on advection.
- Section 3.3: Indeed, it is likely that the cuticular resistance may be overestimated in Massad et al. (2010), as was also found by Loubet et al. (2012) and Personne et al. (2015). However I cannot figure out how to reconcile Fig 10 which shows globally larger deposition fluxes by the model and Fig 11 which shows larger measured deposition fluxes overall. This probably comes from the averaging procedure which splits emissions and deposition in Fig. 10.
- Moreover it would be very instructive here to have comparison of daily dynamics of the modelled and measured flux. This would also allow testing hypothesis with the model, as for instance diminishing  $R_w$ , or adding a compensation point and compare to the observations. Fig. 6 could for instance be duplicated and compared with the model flux or alternatively some example days could be chosen.
- P11L29-30: As mentioned above you should show the inferred  $R_w$  from night time measurements together with  $R_a$  and  $R_b$  or alternatively  $V_{max}(z)$  and  $g_c$ .
- P12L1-2: Please show data from the Delta denuders (in a supplementary section?) or at least give range of concentrations.
- P12L8-9: Please explicit how you extrapolated to the entire year?
- P13L2: You mention in the conclusion the long term stability of the QCL but none was said about it in the manuscript. Either consider withdrawing from the conclusions or add a discussion in the manuscript.

**Tables**

- Table 1. Table and Figure legend should be self-standing: Please explain what are  $c_{\text{NH3}}$ , Ta, P, Rn, SD and what the overbars mean. It may also be useful for interpreting graphs (Fig. 6) or statistics to include the number of points per period.
- Table 2. Explain on what variable was the Kruskal-Wallis test made (the NH3 flux?). It would also be very helpful to do this test on the deposition velocity. But the authors could also consider doing it on the canopy compensation point  $C_c(z_0)$ , the emission potential  $\Gamma(z_0)$  or the canopy conductance  $g_c$ . Pleasealso explicit what are "p-value" and "Post-Hoc" in the legend

**Figures**

- Throughout the text and legends change concentration for mixing ratio.
- Consider adding a Figure with the map of the field and the surrounding including farms and agricultural fields
- Figure 1. It would be helpful to add the heights and the tube length. Also explain or show how are the bypass and "Particles out" channels connected to the pump. Consider also adding the pressure and flow rates on the Scheme. Explicit AC and QCL in the legend.
- Figure 2. I would suggest showing the fitted co-ogive and show explicitly how the frequency damping is evaluated, like in Ammann (2006). Also adding a graph which shows how this frequency damping evolves with time would be very useful. The temperature co-ogive does not see to stabilise completely to 1 at large frequency (we expect from the graph that it may continue growing a bit at larger frequency). Please comment on that in the text.
- Figure 3. This is a very nice graph. It would be helpful to add legend on the right hand side.
- Figure 5. Why splitting the wind direction in periods and not the NH3 mixing ratio wind rose? Here the map of the surrounding would be much needed to help understanding the NH3 wind rose.
- Figure 7. How do you explain the afternoon peak in NH3 mixing ratio. May be I missed it in the text.
- Figure 8. This figure is hard to read. Please consider using lines, a smaller height for the graph and also consider showing additional graphs (as for example in Langford et al. (2009)) of the main drivers ( $u_*$ , Ta, RH, Rn,...). This would ease the discussion and help the reader making his mind on the dataset. Also very important in Figure 8 is to add on the same graph window  $V_{max}(z)$ .
- Figure 9. Consider adding Rn > 20, Rn < 20 and all as legends of the graphs on the graphs themselves. What are the percentage meaning on the top of the graph? Also consider making a similar graph for  $V_d(z)$ .
- Figure 10. This figure needs clarification, and it might be better not to separate emissions and depositions periods. One would expect a compensation point to appear then. Also consider showing half-hourly data instead of pooling. Moreover one would expect pooling to also give horizontal overbars. Also consider showing specific example for one some days with different behaviour : I would expect Period 1 to be like actual Fig 10 left but period three to show a compensation point.
- Figure 11. Before showing daily averages, it would be good to show daily variations. This could be done over a shorter period or using averages as in Figure 6. Consider showing these in the supplementary material.

**References**

- Ammann, C., Brunner, A., Spirig, C., and Neftel, A.: Technical note: Water vapour concentration and flux measurements with PTR-MS, Atmospheric Chemistry and Physics, 6, 4643-4651, 2006.
- Burkhardt, J., Flechard, C. R., Gresens, F., Mattsson, M., Jongejan, P. A. C., Erisman, J. W., Weidinger, T., Meszaros, R., Nemitz, E., and Sutton, M. A.: Modelling the dynamic chemical interactions of atmospheric ammonia with leaf surface wetness in a managed grassland canopy, Biogeosciences, 6, 67-84, 2009.
- Ferrara, R. M., Loubet, B., Di Tommasi, P., Bertolini, T., Magliulo, V., Cellier, P., Eugster, W., and Rana, G.: Eddy covariance measurement of ammonia fluxes: Comparison of high frequency correction methodologies, Agric. For. Meteorol., 158, 30-42, 2012.
- Flechard, C. R., Fowler, D., Sutton, M. A., and Cape, J. N.: A dynamic chemical model of bi-directional ammonia exchange between semi-natural vegetation and the atmosphere, Q.J.R. Meteorol. Soc., 125, 2611-2641, DOI 10.1002/qj.49712555914, 1999.

- Gu, L., Massman, W. J., Leuning, R., Pallardy, S. G., Meyers, T., Hanson, P. J., Riggs, J. S., Hosman, K. P., and Yang, B.: The fundamental equation of eddy covariance and its application in flux measurements, Agric. For. Meteorol., 152, 135-148, 10.1016/j.agrformet.2011.09.014, 2012.
- Hensen, A., Loubet, B., Mosquera, J., van den Bulk, W. C. M., Erisman, J. W., Daemmgen, U., Milford, C., Loepmeier, F. J., Cellier, P., Mikuska, P., and Sutton, M. A.: Estimation of NH3 emissions from a naturally ventilated livestock farm using local-scale atmospheric dispersion modelling, Biogeosciences, 6, 2847-2860, 10.5194/bg-6-2847-2009, 2009.
- Kormann, R., and Meixner, F. X.: An analytical footprint model for non-neutral stratification, Boundary Layer Meteorol., 99, 207-224, 2001.
- Kowalski, A. S., and Serrano-Ortiz, P.: On the relationship between the eddy covariance, the turbulent flux, and surface exchange for a trace gas such as CO2, Boundary Layer Meteorol., 124, 129-141, 10.1007/s10546-007-9171-z, 2007.
- Langford, B., Davison, B., Nemitz, E., and Hewitt, C. N.: Mixing ratios and eddy covariance flux measurements of volatile organic compounds from an urban canopy (Manchester, UK), Atmospheric Chemistry and Physics, 9, 1971-1987, 10.5194/acp-9-1971-2009, 2009.
- Langford, B., Acton, W., Ammann, C., Valach, A., and Nemitz, E.: Eddy-covariance data with low signal-to-noise ratio: time-lag determination, uncertainties and limit of detection, Atmos. Meas. Tech., 8, 4197-4213, 10.5194/amt-8-4197-2015, 2015.
- Loubet, B., Milford, C., Hensen, A., Daemmgen, U., Erisman, J.-W., Cellier, P., and Sutton, M. A.: Advection of NH3 over a pasture field, and its effect on gradient flux measurements, Biogeosciences, 6, 1295-1309, 2009.
- Loubet, B., Decuq, C., Personne, E., Massad, R. S., Flechard, C., Fanucci, O., Mascher, N., Gueudet, J. C., Masson, S., Durand, B., Genermont, S., Fauvel, Y., and Cellier, P.: Investigating the stomatal, cuticular and soil ammonia fluxes over a growing tritical crop under high acidic loads, Biogeosciences, 9, 1537-1552, 10.5194/bg-9-1537-2012, 2012.
- Massad, R. S., Nemitz, E., and Sutton, M. A.: Review and parameterisation of bi-directional ammonia exchange between vegetation and the atmosphere, Atmospheric Chemistry and Physics, 10, 10359-10386, 10.5194/acp-10-10359-2010, 2010.
- Neftel, A., Spirig, C., and Ammann, C.: Application and test of a simple tool for operational footprint evaluations, Environmental Pollution, 152, 644-652, 10.1016/j.envpol.2007.06.062, 2008.
- Personne, E., Tardy, F., Genermont, S., Decuq, C., Gueudet, J. C., MASCHER, N., Durand, B., Masson, S., Lauransot, M., Flechard, C., Burkharlt, J., and Loubet, B.: Investigating sources and sinks for ammonia exchanges between the atmosphere and a wheat canopy following slurry application with trailing hose, Agric. For. Meteorol., 207, 11-23, 10.1016/j.agrformet.2015.03.002, 2015.
- Sutton, M. A., Nemitz, E., Milford, C., Campbell, C., Erisman, J. W., Hensen, A., Cellier, P., David, M., Loubet, B., Personne, E., Schjoerring, J. K., Mattsson, M., Dorsey, J. R., Gallagher, M. W., Horvath, L., Weidinger, T., Meszaros, R., Daemmgen, U., Neftel, A., Herrmann, B., Lehman, B. E., Flechard, C., and Burkhardt, J.: Dynamics of ammonia exchange with cut grassland: synthesis of results and conclusions of the GRAMINAE Integrated Experiment, Biogeosciences, 6, 2907-2934, 10.5194/bg-6-2907-2009, 2009.
- Whitehead, J. D., Twigg, M., Famulari, D., Nemitz, E., Sutton, M. A., Gallagher, M. W., and Fowler, D.: Evaluation of laser absorption spectroscopic techniques for eddy covariance flux measurements of ammonia, Environmental Science & Technology, 42, 2041-2046, 2008.
- Wu, Y. H., Walker, J., Schwede, D., Peters-Lidard, C., Dennis, R., and Robarge, W.: A new model of bi-directional ammonia exchange between the atmosphere and biosphere: Ammonia stomatal compensation point, Agric. For. Meteorol., 149, 263-280, 10.1016/j.agrformet.2008.08.012, 2009.

---

## Referee Comment (RC1) · Anonymous Referee #1 · 19 May 2016

In this study, a closed-path QC-TILDAS system is used to measure bi-directional NH3 fluxes by eddy covariance over peatland. The subject is appropriate for Atmospheric Chemistry and Physics. Given the importance of NH3 deposition with respect to ecosystem processes and a general lack of information on dry deposition, the data are potentially of use to the scientific community with regard to understanding NH3 air-surface exchange in natural ecosystems near local sources (i.e. at high atmospheric NH3 concentrations). The authors note several interesting results which differ from previous studies in natural ecosystems, including the observation of NH3 emissions during wet conditions. While the results are compelling and could be of interest to a broad community of ecologists and atmospheric scientists, there are several issues

noted below that need to be addressed before the paper is acceptable for publication.

Abstract, line 31. The statement regarding the potential for QCL to be applied for NH3 flux measurements within long-term research networks such as NEON may be a bit strong. The concentrations at which the instrument has been applied in this study are not generally representative of NEON sites, in fact they are much higher. The current paper does indeed demonstrate the potential for use at sites influenced by local NH3 sources, where concentrations are relatively high, but further characterization of the instrument performance at low concentrations would be necessary to suggest applicability at sites typical of NEON.

Page 4, line 5. At the flow rate and tubing dimensions noted, was the sample flow fully turbulent? The authors note a high frequency damping factor of 0.67 derived from ogive analysis on page 5 but do not explain the cause. A little more information here would be helpful.

Page 5, Section 2.3. As stated in the introduction, one of the three objectives of the paper is to "test the QCL performance to measure NH3 concentration fluctuations and calculate NH3 fluxes….". In this regard, some additional information on instrument performance and relation to flux calculations is warranted. Specifically, the only mention of precision comes at the top of page 4, line 2, referenced to McManus et al. (2008). A more detailed description of instrument precision would be informative. For example, what is the precision at sampling rates corresponding to frequencies of the flux contributing eddies (see Ferrara et al., 2012)? Related to precision is the flux detection limit. For their setup and site conditions, Ferrara et al (2012) estimated the flux detection limit to be 0.25u*, or about 75ngNH3/m2/s, which is large relative to the fluxes reported in the present study. The authors should include an estimate of flux detection limit in their results.

Page 6, section 2.4. How is the stomatal compensation point parameterized? What value is used for emission potential (gamma)?

Page 7, line 15. "horizontal exchange with higher layers…." Do the authors mean vertical exchange?

Page 7, line 22 – 27. The authors note that they observed low air concentrations at high temperatures in late April/early May. They go on to suggest that this might be related to higher concentrations of acid gases and a tendency of NH3 to shift to the particle phase. Do the DELTA measurements of HCl, SO2, and HNO3 support this statement? Were there particle measurements collected that might help shed some light on this question? The statement regarding volatilization of aerosol in the heated inlet line is a little confusing, as this would tend to bias the measured NH3 high. Some clarification is needed here.

Page 7, discussion of diurnal variability. The authors discuss possible reasons for lower concentrations at night and for the morning increase in concentration. What might be driving the relatively rapid increase during the evening (1600 – 1800) as illustrated in figures 4 and 7?

Page 8, line 20 – 23. The authors mention that emission was observed during rain events. Are the flux measurements valid during active precipitation? Assuming that the measurements are valid, what process would be driving the emission?

Page 9, line 9. ".. the ecosystem emits only under dry conditions in contrary to our observations.." The observation of emission during periods of rain or surface wetness is significant. Are there other published examples where this was observed in natural ecosystems?

Page 10, statistical analysis (Table 2). To me, the statistical analysis does not help explain the patterns of the fluxes. The statistical tests indicate that the groups are different but do they differ in a way that is physically meaningful? What are the results telling the reader about the fluxes? How much of the variability in the fluxes can be explained by their relationships to these variables? If the authors wish to employ statistics to explain the flux patterns, a more well developed and rigorous analysis, which

accounts for collinearity in the independent variables, is needed.

Page 10, line 16. ".. whereas peaks in concentration, although rather forming a bimodal pattern, follow the shape of the air temperature with a 2-3 hour lag". The concentration pattern is very much bimodal, with the evening peak representing a process that is obviously important (as these are mean values) and uncorrelated with temperature. As in my previous comment, some discussion of the cause of this evening mode, which contains the highest concentrations observed diurnally, is warranted. It is too highly amplified and brief to represent boundary layer dynamics. Is it a persistent influence from local sources?

Page 11, line 10. "whereas during emission periods it may be the ammonia flux itself, which is controlling the concentration...". Does this make sense from a mass balance standpoint? Assuming some depth of the boundary layer and no advection, could the measured flux reproduce the observed air concentration?

Page 11, line 12. "...high concentration levels due to advection from local source". Have the authors considered advection as a potential source of error in the flux measurements themselves?

Figure 10. Can the relationship between concentration and flux be used to derive an estimate of the surface compensation point? Regarding the increase in emissions with concentration, it seems a very large compensation point would be needed for the surface to continue emitting at an air concentration of 35 ppb. Some explanation of this feature of the plot would be helpful.

Page 11, line 22. ".. the model does not exhibit any emission". Some additional detail on the model parameterization is needed. What is the stomatal emission potential (gamma) and how was it derived? Does the lack of emission in the model suggest that this parameter should be adjusted?

Page 11, discussion of measured versus modeled fluxes. The results shown in Figure

10 and 11 are difficult to reconcile at first read. The left hand panel of figure 10 shows larger deposition fluxes (more negative) estimated by the model, relative to the measurements, across the entire range of concentrations. The error bars mostly do not overlap. However, figure 11 shows a combination of model overestimation and underestimation of daily mean fluxes across periods. Some explanation of the differences in these plots would be helpful.

Page 11, line 29. The authors note that some analysis of nighttime resistances was conducted which indicated that the Rw produced by the Massad et al scheme is too large for this site. What were the calculated resistances? This seems like a good opportunity to examine why the Massad et al Rw approach is not appropriate for this site. How low should Rw be to optimize the agreement with the measurements? Can Rw and the stomatal emission potential be changed together in a way that better reproduces deposition and emission? If so, does this inform not only the model improvement but also the processes that may be driving the fluxes?

---

## Author Comment (AC1) · 19 Jul 2016

We would like to thank both anonymous referees for their valuable comments on the manuscript. Referee comments are given in bold, the answers in standard font. The first points by Referee 2 were numbered from #1 to #6.

**Answer to Referee 1 (Part A)**

Abstract, line 31. The statement regarding the potential for QCL to be applied for NH3 flux measurements within long-term research networks such as NEON may be a bit strong. The concentrations at which the instrument has been applied in this study are not generally representative of NEON sites, in fact they are much higher. The current paper does indeed demonstrate the potential for use at sites influenced by local NH3 sources, where concentrations are relatively high, but further characterization of the instrument performance at low concentrations would be necessary to suggest applicability at sites typical of NEON.

We will add 'at sites with strong local ammonia sources leading to relatively high mean background concentrations and fluxes.' Indeed we are currently conducting a field campaign at a forest site without local sources and we are able to reliably measure very low concentrations in a range from 0 to 10 ppb.

**Page 4, line 5. At the flow rate and tubing dimensions noted, was the sample flow fully turbulent? The authors note a high frequency damping factor of 0.67 derived from ogive analysis on page 5 but do not explain the cause. A little more information here would be helpful.**

According to the Reynolds number (approx. 2400 - 2700) the flow was not fully turbulent (with Recrit > 3000), but also not laminar. High damping occurs usually because of imperfect turbulent flow regime and not completely excludable adsorption effects in the sampling line. Over time, especially the inlet gets clogged with aerosols, dust etc., which may cause unintended reactions of ammonia. However, due to the relatively short measurement period, we do not expect that increased clogging at the inlet alters the offset of NH3 loss through reactions with particles and wall adsorption effects on the one hand and NH3 production through the volatilization of NH4NO3 aerosols on the other hand as is already discussed on page 4, lines 8–25.

In the EC-community gases like H2O measured with a closed-path device usually undergo spectral damping and are corrected for in a similar way.

Ferrara et al. (2012; 2016), also using an Aerodyne QCL, found similar total damping factors. On Page 5, line 16, we add '..., which usually occurs due to imperfect turbulent flow regime and possible minor wall sorption effects in the sampling line.'

Page 5, Section 2.3. As stated in the introduction, one of the three objectives of the paper is to "test the QCL performance to measure NH3 concentration fluctuations and calculate NH3 fluxes....". In this regard, some additional information on instrument performance and relation to flux calculations is warranted. Specifically, the only mention of precision comes at the top of page 4, line 2, referenced to McManus et al. (2008). A more detailed description of instrument precision would be informative. For example, what is the precision at sampling rates corresponding to frequencies of the flux contributing eddies (see Ferrara et al., 2012)? Related to precision is the flux detection limit. For their setup and site conditions, Ferrara et al (2012) estimated the flux detection limit to be 0.25u\*, or about 75ngNH3/m2/s, which is large relative to the fluxes reported in the present study. The authors should include an estimate of flux detection limit in their results.

We will add information on instrument precision on page 4, line 2*ff*. The main frequencies of the flux contributing eddies were in a range of 0.01 to 1 Hz. Precision, i.e. measurement sensitivity, of the instrument is 0.042, 0.021, 0.016, and 0.010 ppb in 1, 10, 20, and 60 s, respectively. The random flux error was computed according to the method after Finkelstein & Sims (2001) and the corresponding limit of detection was determined after Langford et al. (2015) at a confidence interval of  $\alpha$ =1.96 and resulted in a median of 7.75 ng N m-2 s-1. Alternatively, an upper flux detection limit for half-hourly values can be calculated by using only night-time data under stationary conditions (3 – 7 m s-1 wind speed) and wind from the west-southwest sector, where local ammonia sources were negligible and concentrations were low (< 15 ppb). The standard deviation was 16.5 ng N m-2 s-1, so the 2 $\sigma$ -uncertainty range is 33.0 ng N m-2 s-1. We will add a section about flux detection limits to the manuscript.

**Page 6, section 2.4. How is the stomatal compensation point parameterized? What value is used for emission potential (gamma)?**

We used the N-input dependent parameterization from Eq. (8) of Massad et al. (2010):

$$\Gamma_{\rm s} = 246 + 0.0041 \cdot N_{\rm in}^{3.56}$$

where the annual dry and wet N input to the ecosystem  $N_{in}$  (kg N ha-1 a-1) was estimated to be approximately 25 kg N ha-1 a-1 by Hurkuck et al. (2014) at the same site, leading to a stomatal emission potential  $\Gamma_s$  of 634 mol mol-1.

We will include this information by referring to Massad et al. (2010).

**Page 7, line 15. "horizontal exchange with higher layers . . . ." Do the authors mean vertical exchange?**

Yes, thanks, this was a mistake. Changed to '...vertical exchange with higher layers...'.

Page 7, line 22 – 27. The authors note that they observed low air concentrations at high temperatures in late April/early May. They go on to suggest that this might be related to higher concentrations of acid gases and a tendency of NH3 to shift to the particle phase. Do the DELTA measurements of HCI, SO2, and HNO3 support this statement? Were there particle measurements collected that might help shed some light on this question?

The DELTA measurements do not strongly support this statement as they always cover measurement periods of four weeks. Short occurrences of high concentrations of particles are not identifiable. Hurkuck et al., 2014, who were measuring at weekly time resolution in 2012 and 2013, recorded in the first year especially high HNO3-concentrations in late April and early May, compared to February and March. So there might have been sufficient acid gas species to enhance the conversion from ammonia to ammonium.

**The statement regarding volatilization of aerosol in the heated inlet line is a little confusing, as this would tend to bias the measured NH3 high. Some clarification is needed here.**

It was already mentioned on Page 4 lines 19 - 25, that we possibly overestimate the concentration because of this volatilization effect, but in the case of formation of acidic salts we would regain some of the 'lost' ammonia. This might be a little bit too confusing in this sentence, so we will leave this part out and write: 'With higher temperature a larger amount of acidic gas or particle species are present in the atmosphere, which usually leads to reactions of ammonia to ammonium salts such as ammonium nitrate. (Kim et al., 2011)'

**Page 7, discussion of diurnal variability. The authors discuss possible reasons for lower concentrations at night and for the morning increase in concentration. What might be driving the relatively rapid increase during the evening (1600 – 1800) as illustrated in figures 4 and 7?**

The diurnal concentration pattern is generally highly influenced by the two periods with high concentration between 06 and 15 March and 30 March and 08 April. This can also be observed in Fig. 4, period II and III also exhibit this peak in contrast to the other periods. In these two periods the high concentrations are related to the northeast wind sector, where the next agricultural sources are closest (approx. 1.5 km distance). We separated the diurnal concentration into northeast (0 – 90°) and 90 – 360° wind sectors to illustrate that the late afternoon peak is mainly driven by local ammonia sources located northeast of the tower (see Fig. A1). Fig. A1 will be included in the supplementary material.

Fig. A1: Mean diurnal variation of ammonia concentrations separated by wind direction.

Page 8, line 20 – 23. The authors mention that emission was observed during rain events. Are the flux measurements valid during active precipitation? Assuming that the measurements are valid, what process would be driving the emission? Page 9, line 9. ".. the ecosystem emits only under dry conditions in contrary to our observations.." The observation of emission during periods of rain or surface wetness is significant. Are there other published examples where this was observed in natural ecosystems?

To our knowledge this was the first time it was observed with high-resolution flux measurements. There are emissions as well as depositions during rain events in a range of -75 to 25 ng N m-2 s-1, with a median of -8.6 ng N m-2 s-1. The range is smaller compared to the residual data set, but the median was in a similar magnitude of -14.2 ng N m-2 s-1, so we suppose the data during precipitation to be valid. During precipitation-emission periods temperatures were moderate between 3 and 15 °C, concentrations relatively low between 3 and 11 ppb. The wind direction was mostly west to north, where very few point sources are located. So probably relatively unpolluted air reached the site and led to a gradient between the highly polluted area and the "clean" air, so that some emission occurred.

Page 10, statistical analysis (Table 2). To me, the statistical analysis does not help explain the patterns of the fluxes. The statistical tests indicate that the groups are different but do they differ in a way that is physically meaningful? What are the results telling the reader about the fluxes? How much of the variability in the fluxes can be explained by their relationships to these variables? If the authors wish to employ statistics to explain the flux patterns, a more well developed and rigorous analysis, which accounts for collinearity in the independent variables, is needed.

The statistical analysis was not supposed to explain the patterns of the fluxes as we focus on the methodology and these considerations were only used to show the plausibility of the flux data. We just wanted to know if there are significant influences by the meteorology. The results only tell us, that we cannot exclude the influence of the mentioned meteorological variables. As shown in Fig. 7 (manuscript) especially the diurnal variation of the fluxes and the concentrations appear to be controlled by temperature, net radiation and friction velocity but it is not possible to state to which extend the one or the other value contributes to the flux, because there are highly non-linear processes involved. Only mean-values show this effect because in addition the data is strongly scattered, therewith also some requirements for different statistical tests like for example normal distribution for ANOVA (so we chose the alternative Kruskal-Wallis test) are not met. An in depth analysis would be interesting but we are afraid that this would distract the reader from the main methodological focus of this work.

Page 10, line 16. "... whereas peaks in concentration, although rather forming a bimodal pattern, follow the shape of the air temperature with a 2-3 hour lag". The concentration pattern is very much bimodal, with the evening peak representing a process that is obviously important (as these are mean values) and uncorrelated with temperature. As in my previous comment, some discussion of the cause of this evening mode, which contains the highest concentrations observed diurnally, is warranted. It is too highly amplified and brief to represent boundary layer dynamics. Is it a persistent influence from local sources?

See above. We will add the necessary explanation concerning this peak.

**Page 11, line 10. "whereas during emission periods it may be the ammonia flux itself, which is controlling the concentration . . . ". Does this make sense from a mass balance standpoint? Assuming some depth of the boundary layer and no advection, could the measured flux reproduce the observed air concentration?**

We agree with the reviewer that from a mass balance point of view it might not be applicable, but it was also stated: 'However, more likely is a coincidence of flux drivers and high concentration levels which were high due to advection from the local sources.' Hence, we do not fully apply this statement from Milford et al. (2004) to our site conditions. We will include the following changes for clarification: 'However, Milford's (2004) statement that the concentration may still determine the flux during deposition periods, whereas during emission periods it may be the ammonia flux itself, which is controlling the concentration, is also likely be applicable for our site during deposition periods. In emission periods, a coincidence of flux drivers and high concentration levels, which were high due to advection from the local sources, is the more realistic reason for the relationship in Fig.10 (right panel).'

**Page 11, line 12. "... high concentration levels due to advection from local source". Have the authors considered advection as a potential source of error in the flux measurements themselves?**

Advection affects in the first place concentration and thereby indirectly also the flux. We always had the local sources in mind, that's why we mention it several times in the

manuscript. We will add at page 11 line 15: 'The nearest agricultural ammonia point source was 1.5 km away from the tower. With a measurement height of 2.5 m none of the sources were located within the flux footprint, thus we can largely exclude effects from flux heterogeneity such as a direct contribution of the sources to the measured vertical fluxes. However, there might be still large scale transport processes as outlined in a study by Mohr et al. (2015) that influence ammonia concentrations at the site.'

**Figure 10. Can the relationship between concentration and flux be used to derive an estimate of the surface compensation point? Regarding the increase in emissions with concentration, it seems a very large compensation point would be needed for the surface to continue emitting at an air concentration of 35 ppb. Some explanation of this feature of the plot would be helpful.**

We agree with the reviewer that the site has to have a very large compensation point for these emissions to occur. There are generally two simple ways to derive the canopy compensation point from flux and concentration measurements: (i) Look for situations when the sign of the flux changes from deposition to emission or vice versa and assume the measured air NH3 concentration at these times to be equal to the canopy compensation point, or (ii) invert the single-layer resistance-in-series analogue (details below).

(i) Without separating into emission and deposition and without bin-averaging, the points are too scattered to estimate any compensation point. We tried again to plot the fluxes against the concentration for a smaller flux range (-10 to 10 ng N m-2s-1) without bin-averaging (see figure A2 below), but again we cannot derive a compensation point from it, emission and deposition seems to be divided. Only for single days compensation points appear, but with poor R2. Fig. A2 will be included in the supplementary material.

Fig. A2: Half-hourly scatter plot showing the dependency of NH3 fluxes (only in a range of -10 to 10 ng N m-2s-1) on NH3 concentration, red line: linear regression above for emission, below for deposition, for coefficients and r2 see legend

(ii) We were able to derive a continuous time-series of the canopy compensation point, as proposed by Anonymous Referee 2, using the relation  $\chi_c = F_t \cdot (R_a\{z - d\} + R_b) + \chi_a\{z - d\}$ , where  $\chi_c$  (µg m-3) is the canopy compensation point at the notional height of trace gas

exchange  $z_0'$  (m),  $\chi_a\{z - d\}$  (µg m-3) the air NH3 concentration measured at the aerodynamic reference height z - d (m),  $F_t$  (µg m-2 s-1) is the total NH3 flux measured by the eddy covariance system,  $R_a\{z - d\}$  (s m-1) is the aerodynamic resistance at the reference height, and Rb (s m-1) is the quasi-laminar resistance to NH3 exchange. From this, we were also able to calculate the canopy emission potential as described by Anonymous Referee 2. The results indicate that indeed there appears to be a very large canopy compensation point that closely follows the air NH3 concentration, which triggers emission events and effectively reduces deposition in the way that it prevents NH3 from depositing at the maximum allowed deposition velocity allowed by turbulence ( $v_{d,max} = (R_a + R_b)^{-1}$ ; Fig. A3, panel 1). In other words: In a unidirectional framework, this high canopy compensation point increases the effective canopy resistance Rc (Fig. A3, panel 2), and it appears to have a much larger influence on the observed fluxes than the atmospheric resistances. The constant (stomatal) emission potential from the Massad et al. (2010) model is much lower than the observed canopy emission potentials, and the stomatal compensation point is only a function of temperature, not of the ambient NH3 concentration, which may be an indicator that, at this site, there is another, ambient concentration-dependent bidirectional pathway that is not being modelled (e.g. wet surfaces; as described, for example, by Burkhardt et al. (2009) for the case of leaf wetness).